# CAN WE FAITHFULLY REPRESENT MASKED STATES TO COMPUTE SHAPLEY VALUES ON A DNN?

**Jie Ren, Zhanpeng Zhou, Qirui Chen, Quanshi Zhang**[*]
Shanghai Jiao Tong University

## ABSTRACT

Masking some input variables of a deep neural network (DNN) and computing output changes on the masked input sample represent a typical way to compute attributions of input variables in the sample. People usually mask an input variable using its baseline value. However, there is no theory to examine whether baseline value faithfully represents the absence of an input variable, *i.e.,* removing all signals from the input variable. Fortunately, recent studies (Ren et al., 2023a; Deng et al., 2022a) show that the inference score of a DNN can be strictly disentangled into a set of causal patterns (or concepts) encoded by the DNN. Therefore, we propose to use causal patterns to examine the faithfulness of baseline values. More crucially, it is proven that causal patterns can be explained as the elementary rationale of the Shapley value. Furthermore, we propose a method to learn optimal baseline values, and experimental results have demonstrated its effectiveness.

## 1 INTRODUCTION

Many attribution methods (Zhou et al., 2016; Selvaraju et al., 2017; Lundberg and Lee, 2017; Shrikumar et al., 2017) have been proposed to estimate the attribution (importance) of input variables to the model output, which represents an important direction in explainable AI. In this direction, many studies (Lundberg and Lee, 2017; Ancona et al., 2019; Fong et al., 2019) masked some input variables of a deep neural network (DNN), and they used the change of network outputs on the masked samples to estimate attributions of input variables. As Fig. 1 shows, there are different types of baseline values to represent the absence of input variables.

Theoretically, the trustworthiness of attributions highly depends on whether the current baseline value can really remove the signal of the input variable without bringing in new out-of-distribution (OOD) features. However, there is no criterion to evaluate the signal removal of masking methods. To this end, we need to **first break the *blind faith* that seemingly reasonable baseline values can faithfully represent the absence of input variables, and the blind faith that seemingly OOD baseline values definitely cause abnormal features**. In fact, because a DNN may have complex inference logic, seemingly OOD baseline values do not necessarily generate OOD features.

**Concept/causality-emerging phenomenon.** The core challenge of theoretically guaranteeing or examining whether the baseline value removes all or partial signals of an input variable is to explicitly define the signal/concept/knowledge encoded by a DNN in a countable manner. To this end, Ren et al. (2023a) have discovered a counter-intuitive concept-emerging phenomenon in a trained DNN. Although the DNN does not have a physical unit to encode explicit causality or concepts, Ren et al. (2023a); Deng et al. (2022a) have surprisingly discovered that when the DNN is sufficiently trained, the sparse and symbolic concepts emerge. Thus, we use such concepts as a new perspective to define the optimal baseline value for the absence of input variables.

As Fig. 1 shows, each concept represents an *AND relationship* between a specific set $S$ of input variables. The co-appearance of these input variables makes a numerical contribution $U_S$ to the network output. Thus, we can consider such a concept as a *causal pattern*[1] of the network output,

---

[*]Quanshi Zhang is the corresponding author. He is with the Department of Computer Science and Engineering, the John Hopcroft Center, at the Shanghai Jiao Tong University, China. `zqs1022@sjtu.edu.cn`.

[1]Note that in this paper, the causal pattern means the extracted causal relationship between input variables and the output encoded by the DNN, rather than the true intrinsic causal relationship hidden in data.

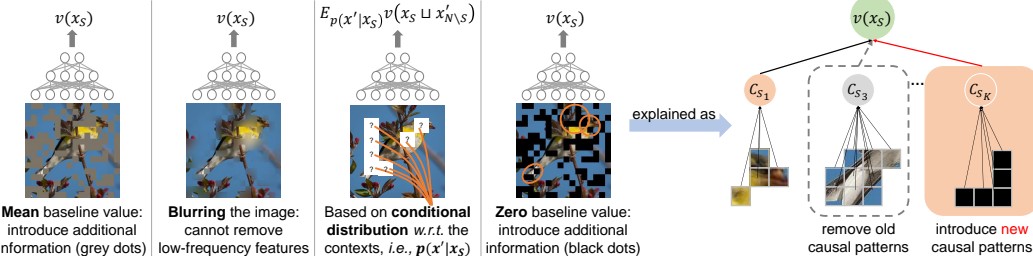

Figure 1: (Left) Previous masking methods may either introduce additional signals, or cannot remove all the old signals. (Right) The inference of the DNN on images masked by these baseline values can be well mimicked by causal patterns.

and $U_S$ is termed the *causal effect*. For example, the concept of a rooster's head consists of the *forehead*, *eyes*, *beak*, and *crown*, *i.e.*, $S = \{forehead, eyes, beak, crown\} = \{f, e, b, c\}$ for short. Only if input variables $f$, $e$, $b$, and $c$ co-appear, the causal pattern $S$ is triggered and makes an effect $U_S$ on the confidence of the head classification. Otherwise, the absence of any input variables in the causal pattern $S$ will remove the effect.

Ren et al. (2023a) have extracted a set of sparse causal patterns (concepts) encoded by the DNN. More importantly, the following finding has proven that *such causal patterns*[1] *can be considered as elementary inference logic used by the DNN*. Specifically, given an input sample with $n$ variables, we can generate $2^n$ different masked samples. *We can use a relatively small number of causal patterns to accurately mimic network outputs on all $2^n$ masked samples, which guarantees the faithfulness of causal patterns.*

**Defining optimal baseline values based on causal patterns.** From the above perspective of causal patterns, whether baseline values look reasonable and fit human's intuition is no longer the key factor to determine the trustworthiness of baseline values. Instead, we evaluate the faithfulness of baseline values by using causal patterns. Because the baseline value is supposed to represent the absence of an input variable, we find that *setting an optimal baseline value usually generates the most simplified explanation of the DNN,* i.e., *we may extract a minimum number of causal patterns to explain the DNN. Such an explanation is the most reliable according to Occam's Razor.*

• We prove that using incorrect baseline values makes a single causal pattern be explained as an exponential number of redundant causal patterns. Let us consider the following toy example, where the DNN contains a causal pattern $S=\{f, e, b, c\}$ with a considerable causal effect $E$ on the output. If an incorrect baseline value $b_f$ of the variable $f$ (*forehead*) just blurs the image patch, rather than fully remove its appearance, then masking the variable $f$ cannot remove all score $E$. The remaining score $E-U_{\{f,e,b,c\}}$ will be explained as redundant causal patterns $U_{\{e,b\}}, U_{\{e,c\}}, U_{\{e,b,c\}}$, *etc.*

• Furthermore, incorrect baseline values may also generate new patterns. For example, if baseline values of $\{f, e, b, c\}$ are set as black regions, then masking all four regions may generate a new pattern of a black square, which is a new causal pattern that influences the network output.

Therefore, we consider that the optimal baseline value, which faithfully reflects the true inference logic, usually simplifies the set of causal patterns. *I.e.,* it usually reduces the overall strength of existing causal effects most without introducing new causal effects. **However, we find that most existing masking methods are not satisfactory from this perspective (see Section 3.2 and Table 1), although the masking method based on conditional distribution of input variables (Covert et al., 2020b; Frye et al., 2021) performs a bit better.**

In particular, we notice that Shapley values can also be derived from causal patterns in theory, *i.e.,* the causal patterns are proven to be elementary effects of Shapley values. Therefore, we propose a new method to learn optimal baseline values for Shapley values, which removes the causal effects of the masked input variables and avoids introducing new causal effects.

**Contributions** of this paper can be summarized as follows. (1) We propose a metric to examine whether the masking approach in attribution methods could faithfully represent the absence state of input variables. Based on this metric, we find that most previous masking methods are not reliable.

(2) We define and develop an approach to estimating optimal baseline values for Shapley values, which ensures the trustworthiness of the attribution.

## 2   EXPLAINABLE AI THEORIES BASED ON GAME-THEORETIC INTERACTIONS

This paper is a typical achievement on the theoretical system of game-theoretic interactions. In fact, our research group has developed and used the game-theoretical interaction as a new perspective to solve two challenges in explainable AI, *i.e., (1) how to define and represent implicit knowledge encoded by a DNN as explicit and countable concepts, (2) how to use concepts encoded by the DNN to explain its representation power or performance.* More importantly, we find that the game-theoretic interaction is also a good perspective to *analyze the common mechanism shared by previous empirical findings and explanations of DNNs.*

● **Explaining the knowledge/concepts encoded by a DNN.** Defining interactions between input variables of a DNN in game theory is a typical research direction (Grabisch and Roubens, 1999; Sundararajan et al., 2020). To this end, we further defined the multi-variate interaction (Zhang et al., 2021a;d) and multi-order interaction (Zhang et al., 2021b) to represent interactions of different complexities. **Ren et al. (2023a) and Li and Zhang (2023) first discovered that we could consider game-theoretic interactions as the concepts encoded by a DNN,** considering the following three terms. **(1)** We found that a trained DNN usually only encoded very sparse and salient interactions, and each interaction made a certain effect on the network output. **(2)** We proved that we could just use the effects of such a small number of salient interactions to well mimic/predict network outputs on an exponential number of arbitrarily masked input samples. **(3)** We found that salient interactions usually exhibited strong transferability across different samples, strong transferability across different DNNs, and strong discrimination power. Thus, **the above three perspectives comprised the solid foundation of considering salient interactions as the concepts encoded by a DNN.** Furthermore, Cheng et al. (2021b) found that such interactions usually represented the most reliable and prototypical concepts encoded by a DNN. Cheng et al. (2021a) further analyzed the different signal-processing behaviors of a DNN in encoding shapes and textures.

● **The game-theoretic interaction is also a new perspective to investigate the representation power of a DNN.** Deng et al. (2022a) proved a counter-intuitive bottleneck/difficulty of a DNN in representing interactions of the intermediate complexity. Zhang et al. (2021b) explored the effects of the dropout operation on interactions to explain the generalization power of a DNN. Wang et al. (2021a;b); Ren et al. (2021) used interactions between input variables to explain the adversarial robustness and adversarial transferability of a DNN. Zhou et al. (2023) found that complex (high-order) interactions were more likely to be over-fitted, and they used the generalization power of different interaction concepts to explain the generalization power of the entire DNN. Ren et al. (2023b) proved that a Bayesian neural network (BNN) was less likely to encode complex (high-order) interactions, which avoided over-fitting.

● **Game-theoretic interactions are also used to analyze the common mechanism shared by many empirical findings.** Deng et al. (2022b) discovered that almost all (fourteen) attribution methods could be re-formulated as a reallocation of interactions in mathematics. This enabled the fair comparison between different attribution methods. Zhang et al. (2022) proved that twelve previous empirical methods of boosting adversarial transferability could be explained as reducing interactions between pixel-wise adversarial perturbations.

## 3   PROBLEMS WITH THE REPRESENTATION OF THE MASKED STATES

**The Shapley value** (Shapley, 1953) was first introduced in game theory to measure the contribution of each player in a game. People usually use Shapley values to estimate attributions of input variables of a DNN. Let the input sample $\boldsymbol{x}$ of the DNN contain $n$ input variables, *i.e.,* $\boldsymbol{x} = [x_1, \ldots, x_n]$. The Shapley value of the $i$-th input variable $\phi_i$ is defined as follows.

$$\phi_i = \sum\nolimits_{S \subseteq N \setminus \{i\}} [|S|!(n - |S| - 1)!/n!] \cdot [v(\boldsymbol{x}_{S \cup \{i\}}) - v(\boldsymbol{x}_S)] \tag{1}$$

where $v(\boldsymbol{x}_S) \in \mathbb{R}$ denotes the model output when variables in $S$ are present, and variables in $N \setminus S$ are masked. Specifically, $v(\boldsymbol{x}_\emptyset)$ represents the model output when all input variables are masked. The Shapley value of the variable $i$ is computed as the weighted marginal contribution of $i$ when the variable $i$ is present *w.r.t.* the case when the variable $i$ is masked, *i.e.* $v(\boldsymbol{x}_{S \cup \{i\}}) - v(\boldsymbol{x}_S)$.

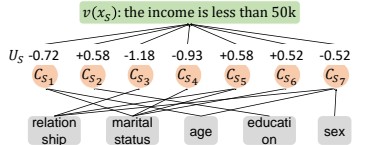

Figure 2: Causal patterns that explain the inference on a sample in the income dataset.

Table 1: The ratio $R$ of the remaining and newly introduced causal effects in the masked inputs. A small value of $R$ meant that baseline values removed most original causal effects and did not introduce many new effects.

|  | $R^{\text{(zero)}}$ | $R^{\text{(mean)}}$ | $R^{\text{(blur)}}$ | $R^{\text{(conditional)}}$ | $R^{\text{(ours)}}$ |
|---|---|---|---|---|---|
| MNIST | 1.1736 | 0.3043 | 0.4159 | 0.3780 | **0.2185** |
| CIFAR-10 | 0.6630 | 0.8042 | 0.7288 | 0.4027 | **0.1211** |

The Shapley value is widely considered a fair attribution method, because it satisfies the *linearity, dummy, symmetry*, and *efficiency* axioms (Weber, 1988) (please refer to Appendix D). However, when we explain a DNN, a typical challenge is how to faithfully define the absence of an input variable. The most classical way is to use ***baseline values*** (or called *reference values*) $\boldsymbol{b} = [b_1, b_2, \ldots, b_n]$ to mask variables to represent their absence. Specifically, given an input sample $\boldsymbol{x}$, $\boldsymbol{x}_S$ denotes a masked sample, which is generated by masking variables in the set $N \setminus S$.

$$\text{If } i \in S, \ (\boldsymbol{x}_S)_i = x_i; \ \text{ otherwise, } (\boldsymbol{x}_S)_i = b_i \tag{2}$$

We aim to learn optimal baseline values $\boldsymbol{b}$ to faithfully represent absent states of input variables.

**Decomposing a DNN's output into sparse interactions.** Given a trained DNN $v$ and an input $\boldsymbol{x}$ with $n$ input variables, Ren et al. (2023a) have proven that the DNN output $v(\boldsymbol{x})$ can be decomposed into effects of interactions between input variables. Specifically, let $S \subseteq N$ denote a subset of input variables. The interaction effect between variables in $S$ is defined as the following Harsanyi dividend (Harsanyi, 1982).

$$U_S \overset{\text{def}}{=} \sum_{S' \subseteq S} (-1)^{|S| - |S'|} \cdot v(\boldsymbol{x}_{S'}) \tag{3}$$

Based on this definition, we have $v(\boldsymbol{x}) = \sum_{S \subseteq N} U_S$.

**Sparse salient interactions can be considered as causal patterns[1] (or concepts) encoded by the DNN.** Theorem 1 and Remark 1 prove that most interactions have ignorable effects $U_S \approx 0$, and people can use a few salient interactions with non-ignorable effects to well approximate the inference scores on $2^n$ different masked samples. Thus, we can consider such interactions as causal patterns[1] or concepts encoded by the DNN. Accordingly, we can consider the interaction effect $U_S$ as the causal effect. Besides, Remark 1 has been verified on different DNNs learned for various tasks by experiments in both Appendix G.1 and (Ren et al., 2023a).

**Theorem 1** (***Faithfulness**, proven by Ren et al. (2023a) and Appendix E.1) Let us consider a DNN $v$ and an input sample $\boldsymbol{x}$ with $n$ input variables. We can generate $2^n$ different masked samples, i.e., $\{\boldsymbol{x}_S | S \subseteq N\}$. The DNN's outputs on all masked samples can always be well mimicked as the sum of the triggered interaction effects in Eq. (3), i.e., $\forall S \subseteq N, v(\boldsymbol{x}_S) = \sum_{S' \subseteq S} U_{S'}$.*

**Remark 1** (*Sparsity*) *Interaction effects in most DNNs are usually very sparse. Most interaction effects are almost zero, i.e., $U_S \approx 0$. A few most salient interaction effects in $\Omega$ (less than 100 interaction effects in most cases) are already enough to approximate the DNN, i.e., $\forall S \subseteq N, v(\boldsymbol{x}_S) \approx \sum_{S' \in \Omega, S' \subseteq S} U_{S'}$, where $|\Omega| \ll 2^n$.*

Each causal pattern (concept) $S$ represents an AND relationship between input variables in $S$. For example, the head pattern of a rooster consists of $\{forehead, eyes, beak, crown\}$. If the *forehead*, *eyes*, *beak*, and *crown* of the rooster co-appear, then the head pattern $S = \{forehead, eyes, beak, crown\}$ is triggered and makes a causal effect $U_S$ on the output. Otherwise, if any part is masked, the causal pattern $S$ will not be triggered, and the DNN's inference score $v(\boldsymbol{x})$ will not receive the causal effect $U_S$. In sum, when we mask an input variable $i$, it is supposed to remove all causal effects of all AND relationships that contain the variable $i$. Please see Appendix F for the proof.

### 3.1 Examining the faithfulness of baseline values using causal patterns

We use salient causal patterns (or concepts) to evaluate the faithfulness of masking methods. Specifically, we examine *whether baseline values remove most causal effects depending on $x_i$, and whether baseline values generate new causal effects.*

The evaluation of the masking methods based on salient causal effects is theoretically supported from the following three perspectives. First, Theorem 1 and Remark 1 prove that the inference score of a DNN can be faithfully disentangled into a relatively small number of causal patterns. Second, Theorem 2 shows that Shapley values can be explained as a re-allocation of causal effects to input

variables. Therefore, *reducing effects of salient patterns means the removal of elementary factors that determine Shapley values.* Besides, in order to verify that the reduction of causal patterns can really represent the absence of input variables, we have conducted experiments to find that *salient patterns triggered by white noise inputs were much less than those triggered by normal images.* Please see Appendix G.2 for details.

**Theorem 2** *(proven by Harsanyi (1982) and Appendix E.2) We can directly derive Shapley values from the effects $U_S$ of causal patterns. The Shapley value can be considered as uniformly allocating each causal pattern $S$'s effect $U_S$ to all its variables,* i.e. $\phi_i = \sum_{S \subseteq N \setminus \{i\}} \frac{1}{|S|+1} U_{S \cup \{i\}}$.

Third, an incorrect baseline value $b_i$ will make partial effects of the AND relationship of the variable $i$ be mistakenly explained as an exponential number of additional redundant causal patterns, which significantly complicates the explanation. Therefore, *the optimal baseline value is supposed to generate the most sparse causal patterns as the simplest explanation of the DNN. Compared to dense causal patterns generated by sub-optimal baseline values, the simplest explanation removes as many as existing causal effects as possible without introducing additional causal effects.*

**Remark 2** *(proof in Appendix E.3) Let us consider a function with a single causal pattern $f(\boldsymbol{x}_S) = w_S \prod_{j \in S} (x_j - \delta_j)$. Accordingly, ground-truth baseline values of variables are obviously $\{\delta_j\}$, because setting any variable $\forall j \in S$, $x_j = \delta_j$ will deactivate this pattern. Given the correct baseline values $b_j^* = \delta_j$, we can use a single causal pattern to regress $f(\boldsymbol{x}_S)$, i.e., $U_S = f(\boldsymbol{x}_S)$, $\forall S' \neq S, U_{S'} = 0$.*

**Theorem 3** *(proof in Appendix E.3) For the function $f(\boldsymbol{x}_S) = w_S \prod_{j \in S} (x_j - \delta_j)$, if we use $m'$ incorrect baseline values $\{b_j' | b_j' \neq \delta_j\}$ to replace correct ones to compute causal effects, then the function will be explained to contain at most $2^{m'}$ causal patterns.*

**Theorem 4** *(proof in Appendix E.3) If we use $m'$ incorrect baseline values to compute causal effects in the function $f(\boldsymbol{x}_S) = w_S \prod_{j \in S} (x_j - \delta_j)$, a total of $\binom{m'}{k - |S| + m'}$ causal patterns of the $k$-th order emerge, $k \geq |S| - m'$. A causal pattern of the $k$-th order means that this causal pattern represents the AND relationship between $k$ variables.*

Specifically, Remark 2, Theorems 3 and 4 provide a new perspective to understand how incorrect baseline values generate new causal patterns. Remark 2 shows how correct baseline values explain a toy model that contains a single causal pattern. **Theorems 3 and 4 show that incorrect baseline values will use an exponential number of redundant low-order patterns to explain a single high-order causal pattern.** For example, we are given the function $f(\boldsymbol{x}) = w(x_p - \delta_p)(x_q - \delta_q)$ s.t. $x_p = 3, x_q = 4, \delta_p = 2, \delta_q = 3$. If we use ground-truth baseline values $\{\delta_p, \delta_q\}$, then the function is explained as simple as a single causal pattern $\Omega = \{\{p, q\}\}$, which yields correct Shapley values $\phi_p = \phi_q = 0.5 \cdot w$, according to Theorem 2. Otherwise, if we use incorrect baseline values $\{b_p' = 1, b_q' = 1\}$, then this function will be explained as four causal patterns $\Omega = \{\emptyset, \{p\}, \{q\}, \{p, q\}\}$, *i.e.,* $f(\boldsymbol{x}) = U_\emptyset C_\emptyset + U_{\{p\}} C_{\{p\}} + U_{\{q\}} C_{\{q\}} + U_{\{p,q\}} C_{\{p,q\}}$, where $U_\emptyset = 2w$, $U_{\{p\}} = -4w$, $U_{\{q\}} = -3w$, and $U_{\{p,q\}} = 6w$ are computed using incorrect baseline values. Incorrect baseline values increase complicated causal patterns and lead to incorrect Shapley values $\phi_p = -w, \phi_q = 0$. In fact, the existence of most newly introduced causal patterns is due to that the effects of a high-order causal pattern are not fully removed, and that OOD causal patterns (new OOD edges or shapes) may be caused by incorrect baseline values.

## 3.2 Problems with previous masking methods

In this subsection, we compare causal patterns in the masked sample with causal patterns in the original sample to evaluate the following baseline values.

(1) *Mean baseline values.* As Fig. 1 shows, the baseline value of each input variable is set to the mean value of this variable over all samples (Dabkowski and Gal, 2017), *i.e.* $b_i = \mathbb{E}_x[x_i]$. However, empirically, this method actually introduces additional signals to the input. For example, mean values introduce massive grey dots to images and may form new edges as abnormal causal patterns. This has been verified by experiments in Table 1. Experimental details will be introduced later.

(2) *Zero baseline values.* Baseline values of all input variables are set to zero (Ancona et al., 2019; Sundararajan et al., 2017), *i.e.* $\forall i \in N, b_i = 0$. As Fig. 1 shows, just like mean baseline values, zero baseline values also introduce additional signals (black dots) to the input (verified in Table 1).

Table 2: Analysis about previous masking methods.

| Setting of baseline values | Baseline values are constant or not | Different samples share the same baseline values or not | Shortcomings |
|---|---|---|---|
| Mean baseline value | ✓ | ✓ | introduce additional signals |
| Zero baseline value | ✓ | ✓ | introduce additional signals |
| Blurring baseline value | ✓ | ✗ | not remove low-frequency components |
| Marginal distribution | ✗ | ✓ | assume feature independence |
| Conditional distribution | ✗ | ✗ | balance between the computational cost and the accuracy |
| Ours | ✓ | both are ok | high computational cost |

(3) *Blurring input samples.* Fong and Vedaldi (2017) and Fong et al. (2019) blur image pixels $x_i$ using a Gaussian kernel as its masked state. Covert et al. (2020a); Sturmfels et al. (2020) mentioned that this approach only removed high-frequency signals, but failed to remove low-frequency signals.

(4) *For each input variable, determining a different baseline value for each specific context $S$.* Instead of fixing baseline values as constants, some studies use varying baseline values to compute $v(\boldsymbol{x}_S)$ given $\boldsymbol{x}$, which are determined temporarily by the context $S$ in $\boldsymbol{x}$. Some methods (Frye et al., 2021; Covert et al., 2020b) define $v(\boldsymbol{x}_S)$ by modeling the conditional distribution of variable values in $N \setminus S$ given the context $S$, *i.e.* $v(\boldsymbol{x}_S) = \mathbb{E}_{p(\boldsymbol{x}'|\boldsymbol{x}_S)}[model(\boldsymbol{x}_S \sqcup \boldsymbol{x}'_{N\setminus S})]$. The operation $\sqcup$ means the concatenation of $\boldsymbol{x}$'s dimensions in $S$ and $\boldsymbol{x}'$'s dimensions in $N \setminus S$. By assuming the independence between input variables, the above conditional baseline values can be simplified to marginal baseline values (Lundberg and Lee, 2017), *i.e.* $v(\boldsymbol{x}_S) = \mathbb{E}_{p(\boldsymbol{x}')}[model(\boldsymbol{x}_S \sqcup \boldsymbol{x}'_{N\setminus S})]$.

We conducted experiments to examine whether the above baseline values remove all causal patterns in the original input and whether baseline values introduce new causal patterns. We used the metric $R = \mathbb{E}_{\boldsymbol{x}}\Big[(\sum_{S\subseteq N} |U'_S| - \sum_{S\subseteq N} |U_S^{(\text{noise})}|)/(\sum_{S\subseteq N} |U_S|)\Big]$ to evaluate the quality of masking. We generated a set of samples based on $\boldsymbol{x}$, where a set of input variables were masked, and $U'_S$ denote the causal effect in such masked samples. $U_S$ denote the causal effect in the original sample $\boldsymbol{x}$, which was used for normalization. $U_S^{(\text{noise})}$ denotes the causal effect in a white noise input, and it represents the unavoidable effect of huge amounts of noise patterns. Thus, we considered the $U_S^{(\text{noise})}$ term as an inevitable anchor value and removed it from $R$ for a more convincing evaluation. The masking method would have two kinds of effects on causal patterns. (1) We hoped to remove all existing salient patterns in the original sample. (2) We did not expect the masking method to introduce new salient patterns. Interestingly, the removal of existing salient patterns decreased the $R$ value, while the triggering of new patterns increased the $R$ value. Thus, the $R$ metric reflected both effects. A small value of $R$ indicated a good setting of baseline values.

We used 20 images in the MNIST dataset (LeCun et al., 1998) and 20 images in the CIFAR-10 dataset (Krizhevsky et al., 2009) to compute $R$, respectively. We split each MNIST image into $7 \times 7$ grids and split each CIFAR-10 image into $8 \times 8$ grids. For each image, we masked the central $4 \times 3$ grids using the zero baseline, mean baseline, blur baseline, and the baseline based on the conditional distribution, and computed the metric of $R^{(\text{zero})}$, $R^{(\text{mean})}$, $R^{(\text{blur})}$, and $R^{(\text{conditional})}$, respectively. Table 1 shows that the ratio $R$ by using previous baseline values were all large. Although the masking method based on conditional distribution performed better than some other baseline values, our method exhibited the best performance. **It indicates that previous masking methods did not remove most existing patterns and/or trigger new patterns.**

### 3.3 Absence states and optimal baseline values

In the original scenario of game theory, the Shapley value was proposed without the need to define the absence of players. When people explain a DNN, we consider that the **true absence state** of variables should generate the most simplified causal explanation. Remark 2 and Theorem 3 show that correct baseline values usually generate the simplest causal explanation, *i.e.,* using the least number of causal patterns to explain the DNN. In comparison, if an incorrect baseline value $b_i$ does not fully remove all effects of AND relationships of the variable $i$, then the remained effects will be mistakenly explained as a large number of other redundant patterns.

The above proof well fits Occam's Razor, *i.e.*, the simplest causality with the minimum causal patterns is more likely to represent the essence of the DNN's inference logic. This also lets us consider the baseline values that minimize the number of salient causal patterns (*i.e.*, achieving the simplest causality) as the optimal baseline values.

Therefore, the learning of the baseline value $b_i^*$ of the $i$-th variable can be formulated to sparsify causal patterns in the deep model. Particularly, such baseline values are supposed to remove existing

Table 3: Examples of generated functions and their ground-truth baseline values.

| Functions ($\forall i \in N, x_i \in \{0,1\}$) | The ground truth of baseline values |
|---|---|
| $-0.185x_1(x_2 + x_3)^{2.432} - x_4x_5x_6x_7$ | $b_i^* = 0$ for $i \in \{1,2,3,4,5,6,7\}$ |
| $-x_1x_2x_3 + sigmoid(-5x_4x_5x_6x_7 + 2.50) - x_8x_9$ | $b_i^* = 1$ for $i \in \{4,5,6,7\}$, $b_i^* = 0$ for $i \in \{1,2,3,8,9\}$ |
| $-sigmoid(+4x_1 - 4x_2 + 4x_3 - 6.00) - x_4x_5x_6x_7 - x_8x_9x_{10}$ | $b_i^* = 1$ for $i = 2$, $b_i^* = 0$ for $i \in \{1,3,4,5,6,7,8,9,10\}$ |

causal effects without introducing many new effects.

$$\boldsymbol{b}^* = \arg\min_{\boldsymbol{b}} \sum_{\boldsymbol{x}} |\Omega(\boldsymbol{x})|, \quad \text{subject to } \Omega(\boldsymbol{x}) = \{S \subseteq N \mid |U_S(\boldsymbol{x}|\boldsymbol{b})| > \tau\} \tag{4}$$

where $U_S(\boldsymbol{x}|\boldsymbol{b})$ denotes the causal effect computed on the sample $\boldsymbol{x}$ by setting baseline values to $\boldsymbol{b}$.

## 4 ESTIMATING BASELINE VALUES

Based on Theorem 3, we derive Eq. (4) to learn optimal baseline values, but the computational cost of enumerating all causal patterns is exponential. Thus, we explore an approximate solution to learning baseline values. According to Theorem 4, *incorrect baseline values usually mistakenly explain high-order causal patterns as an unnecessarily large number of low-order causal patterns,* where the order $m$ of the causal effect $U_S$ is defined as the cardinality of $S$, $m = |S|$.

Thus, *the objective of learning baseline values* is roughly equivalent to penalizing effects of low-order causal patterns, in order to prevent learning incorrect baseline values that mistakenly represent the high-order pattern as an exponential number of low-order patterns.

$$\min_{\boldsymbol{b}} L(\boldsymbol{b}), \quad \text{subject to } L(\boldsymbol{b}) = \sum_{\boldsymbol{x}} \sum_{S \subseteq N, |S| \le k} |U_S(\boldsymbol{x}|\boldsymbol{b})| \tag{5}$$

**An approximate-yet-efficient solution.** When each input sample contains a huge number of variables, *e.g.*, an image sample, directly optimizing Eq. (5) is NP-hard. Fortunately, we find the multi-order Shapley value and the multi-order marginal benefit in the following equation have strong connections with multi-order causal patterns (proven in Appendix H), as follows.

$$\phi_i^{(m)}(\boldsymbol{x}|\boldsymbol{b}) \overset{\text{def}}{=} \mathbb{E}_{\substack{S \subseteq N \setminus \{i\} \\ |S|=m}} \left[ v(\boldsymbol{x}_{S \cup \{i\}}, \boldsymbol{b}) - v(\boldsymbol{x}_S, \boldsymbol{b}) \right] = \mathbb{E}_{\substack{S \subseteq N \setminus \{i\} \\ |S|=m}} \left[ \sum_{L \subseteq S} U_{L \cup \{i\}}(\boldsymbol{x}|\boldsymbol{b}) \right]$$

$$\Delta v_i(S|\boldsymbol{x}, \boldsymbol{b}) \overset{\text{def}}{=} v(\boldsymbol{x}_{S \cup \{i\}}, \boldsymbol{b}) - v(\boldsymbol{x}_S, \boldsymbol{b}) = \sum_{L \subseteq S} U_{L \cup \{i\}}(\boldsymbol{x}|\boldsymbol{b}) \tag{6}$$

where $\phi_i^{(m)}(\boldsymbol{x}|\boldsymbol{b})$ and $\Delta v_i(S|\boldsymbol{x}, \boldsymbol{b})$ denote the $m$-order Shapley value and the $m$-order marginal benefit computed using baseline values $\boldsymbol{b}$, respectively, where the order $m$ is given as $m = |S|$.

According to the above equation, high-order casual patterns $U_S$ are only contained by high-order Shapley values $\phi_i^{(m)}$ and high-order marginal benefits $\Delta v_i$. Therefore, in order to penalize the effects of low-order causal patterns, we penalize the strength of low-order Shapley values and low-order marginal benefits, respectively, as an engineering solution to boost computational efficiency. In experiments, these loss functions were optimized via SGD.

$$L_{\text{Shapley}}(\boldsymbol{b}) = \sum_{m \sim \text{Unif}(0,\lambda)} \sum_{\boldsymbol{x} \in X} \sum_{i \in N} |\phi_i^{(m)}(\boldsymbol{x}|\boldsymbol{b})|, \quad L_{\text{marginal}}(\boldsymbol{b}) = \sum_{m \sim \text{Unif}(0,\lambda)} \sum_{\boldsymbol{x} \in X} \sum_{i \in N} \mathbb{E}_{\substack{S \subseteq N \\ |S|=m}} |\Delta v_i(S|\boldsymbol{x}, \boldsymbol{b})| \tag{7}$$

where $\lambda \ge m$ denotes the maximum order to be penalized. We have conducted experiments to verify that baseline values $\boldsymbol{b}$ learned by loss functions in Eq. (7) could effectively sparsify causal effects of low-order causal patterns in Eq. (5). Please see Appendix G.3 for results.

*Most importantly, we still used the metric $R$ in Section 3.2 to check whether the learned baseline values removed original causal patterns in the input while not introducing new patterns.* The low value of $R^{\text{(ours)}}$ in Table 1 shows that baseline values learned by our method **successfully removed existing salient causal patterns without introducing many new salient patterns.**

## 5 EXPERIMENTS

### 5.1 VERIFICATION OF CORRECTNESS OF BASELINE VALUES AND SHAPLEY VALUES

**Correctness of baseline values on synthetic functions.** People usually cannot determine the ground truth of baseline values for real images, such as the MNIST dataset. Therefore, we conducted experiments on synthetic functions with ground-truth baseline values, in order to verify the

Table 4: Accuracy of the learned baseline values.

|  | $L_{\text{Shapley}}$ | | | $L_{\text{marginal}}$ | | |
|---|---|---|---|---|---|---|
|  | initialize as 0 | initialize as 0.5 | initialize as 1 | initialize as 0 | initialize as 0.5 | initialize as 1 |
| Synthetic functions | 98.06% | 98.70% | 98.70% | 98.06% | 98.14% | 98.14% |
| Functions in (Tsang et al., 2018) | 88.52% | 91.80% | 90.16% | 86.89% | 91.80% | 90.16% |

Table 5: Accuracy of Shapley values on the extended Addition-Multiplication dataset using different settings of baseline values.

|  | Zero | Mean | Baseline values in SHAP | Kernel SHAP | Frye et al. (2021) | Ours |
|---|---|---|---|---|---|---|
| Accuracy | 82.88% | 72.63% | 81.25% | 33.88% | 66.00% | 100% |

Table 6: An example of Shapley values computed on different baseline values. The function is $model(\boldsymbol{x}) = -2.62x_1 - 5x_3 - 1.98x_6(x_4 - 0.94) + 1.15(x_5 - 0.91) - 4.23x_7$, and the input is $\boldsymbol{x} = [0, 1, 1, 1, 1, 1, 1]$.

| Baseline values | The computed Shapley values $\{\phi_i\}$ |
|---|---|
| Truth/Ours | $\{0, 0, -5, -0.06, 0.10, -0.06, -4.23\}$ |
| Zero baseline | $\{0, 0, -5, -0.99, 1.15, 0.87, -4.23\}$ |
| Mean baseline | $\{1.31, 0, -2.50, -0.74, 0.58, 0.19, -2.11\}$ |
| Setting in SHAP | $\{0.014, 0, -0.011, -0.003, 0.003, 0.001, -0.010\}$ |

correctness of the learned baseline values. We randomly generated 100 functions, whose causal patterns and ground truth of baseline values could be easily determined. This dataset has been released at `https://github.com/zzp1012/faithful-baseline-value`. The generated functions were composed of addition, subtraction, multiplication, exponentiation, and the *sigmoid* operations (see Table 3). For example, for the function $y = sigmoid(3x_1x_2 - 3x_3 - 1.5) - x_4x_5 + 0.25(x_6 + x_7)^2$, $x_i \in \{0, 1\}$, there were three causal patterns (*i.e.* $\{x_1, x_2, x_3\}, \{x_4, x_5\}, \{x_6, x_7\}$), which were activated only if $x_i = 1$ for $i \in \{1, 2, 4, 5, 6, 7\}$ and $x_3 = 0$. In this case, the ground truth of baseline values was $b_i^* = 0$ for $i \in \{1, 2, 4, 5, 6, 7\}$ and $b_3^* = 1$. Please see Appendix G.4 for more discussions about the setting of ground-truth baseline values.

We used our method to learn baseline values on these functions and tested the accuracy. Note that $|b_i - b_i^*| \in [0, 1]$ and $b_i^* \in \{0, 1\}$. If $|b_i - b_i^*| < 0.5$, we considered the learned baseline value correct. We set $\lambda = 0.5n$ in both $L_{\text{Shapley}}$ and $L_{\text{marginal}}$. The results are reported in Table 4 and are discussed later.

**Correctness of baseline values on functions in (Tsang et al., 2018).** Besides, we also evaluated the correctness of the learned baseline values using functions in Tsang et al. (2018). Among all the 92 input variables in these functions, the ground truth of 61 variables could be determined (see Appendix G.4). Thus, we used these annotated baseline values to test the accuracy. Table 4 reports the accuracy of the learned baseline values on the above functions. In most cases, the accuracy was above 90%, showing that our method could effectively learn correct baseline values. A few functions in (Tsang et al., 2018) did not have salient causal patterns, which caused errors in the learning. Besides, in experiments, we tested our method under three different initializations of baseline values (*i.e.,* 0, 0.5, and 1). Table 4 shows that baseline values learned with different initialization settings all converged to similar and high accuracy.

**Correctness of the computed Shapley values. Incorrect baseline values lead to incorrect Shapley values.** We verified the correctness of the computed Shapley values on the extended Addition-Multiplication dataset (Zhang et al., 2021c). We added the subtraction operation to avoid all baseline values being zero. Theorem 2 considers the Shapley value as a uniform assignment of effects of each causal pattern to its compositional variables. This enabled us to determine the ground-truth Shapley value of variables without baseline values based on causal patterns. For example, the function $f(\boldsymbol{x}) = 3x_1x_2 + 5x_3x_4 + x_5$ *s.t.* $\boldsymbol{x} = [1, 1, 1, 1, 1]$ contained three causal patterns, according to the principle of the most simplified causality. Accordingly, the ground-truth Shapley values were $\hat{\phi}_1 = \hat{\phi}_2 = 3/2$, $\hat{\phi}_3 = \hat{\phi}_4 = 5/2$, and $\hat{\phi}_5 = 1$. See Appendix G.5 for more details. The estimated Shapley value $\phi_i$ was considered correct if $|\phi_i - \hat{\phi}_i| \leq 0.01$; otherwise, incorrect. Then, we computed the accuracy of the estimated Shapley values as the ratio of input variables with correct Shapley values.

*Discussion on why the learned baseline values generated correct Shapley values.* We computed Shapley values of variables in the extended Addition-Multiplication dataset using different baseline values, and compared their accuracy in Table 5. The result shows that our method exhibited the highest accuracy. Table 6 shows an example of incorrect Shapley values computed by using other baseline values. Our method generated correct Shapley values in this example. For the variable $x_6$, due to its negative coefficient $-1.98$, its contribution should be negative. However, all other baseline values generated positive Shapley values for $x_6$. The term $-4.23x_7$ showed the significant effect of the variable $x_7$ on the output, but its Shapley value computed using baseline values in SHAP was just $-0.010$, which was obviously incorrect.

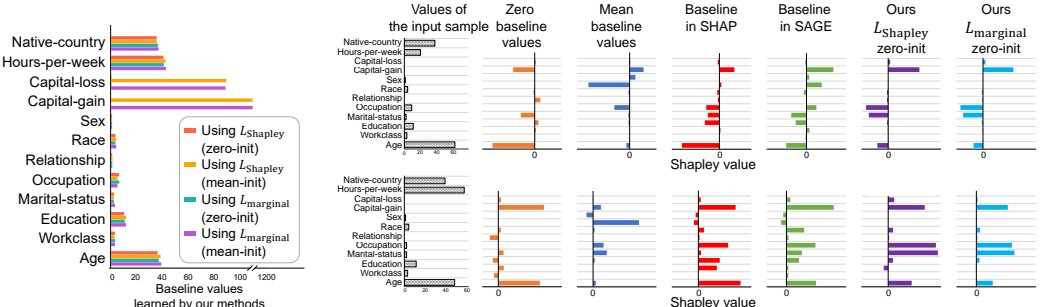

Figure 3: The learned baseline values (left) and Shapley values computed with different baseline values (right) on the income dataset. **Results on the MNIST, the CIFAR-10, and the credit datasets are shown in Appendix G.6 and G.7.**

## 5.2 RESULTS AND EVALUATION ON REALISTIC DATASETS AND MODELS

**Learning baseline values.** We used our method to learn baseline values for MLPs, LeNet (Le-Cun et al., 1998), and ResNet-20 (He et al., 2016) trained on the UCI South German Credit dataset (namely *credit dataset*) (Dua and Graff, 2017), the UCI Census Income dataset (namely *income dataset*) (Dua and Graff, 2017), the MNIST dataset (LeCun et al., 1998), and the CIFAR-10 dataset (Krizhevsky et al., 2009), respectively. We learned baseline values by using either $L_{\text{Shapley}}$ or $L_{\text{marginal}}$ as the loss function. In the computation of $L_{\text{Shapley}}$, we set $v(\boldsymbol{x}_S) = \log \frac{p(y^{\text{truth}}|\boldsymbol{x}_S)}{1-p(y^{\text{truth}}|\boldsymbol{x}_S)}$. In the computation of $L_{\text{marginal}}$, $|\Delta v_i(S)|$ was set to $|\Delta v_i(S)| = \|h(\boldsymbol{x}_{S \cup \{i\}}) - h(\boldsymbol{x}_S)\|_1$, where $h(\boldsymbol{x}_S)$ denotes the output feature of the penultimate layer given the masked input $\boldsymbol{x}_S$, in order to boost the efficiency of learning. We set $\lambda = 0.2n$ for the MNIST and the CIFAR-10 datasets, and set $\lambda = 0.5n$ for the simpler data in two UCI datasets. Given baseline values, we used the sampling-based approximation (Castro et al., 2009) to estimate Shapley values.

We used two ways to initialize baseline values before learning, *i.e.* setting baseline values to zero or mean values over different samples, namely *zero-init* and *mean-init*, respectively. Fig. 3 (left) shows that baseline values learned with different initialization settings all converged to similar baseline values, except for very few dimensions having multiple local-minimum solutions (discussed in Appendix G.7), which proved the stability of our method.

**Comparison of attributions computed using different baseline values.** Fig. 3 shows the learned baseline values and the computed Shapley values on the income dataset. We found that attributions generated by zero/mean baseline values conflicted with the results of all other methods. Our method obtained that the *occupation* had more influence than the *marital status* on the income, which was somewhat consistent with our life experience. However, baseline values in SHAP and SAGE sometimes generated abnormal explanations. In this top-right example, the attribute *capital gain* was zero, which was not supposed to support the prediction of "the person made over 50K a year." However, the SAGE's baseline values generated a large positive Shapley value for *capital gain*. In the bottom-right example, both SHAP and SAGE considered the *marital status* important for the prediction. SHAP did not consider the *occupation* as an important variable. Therefore, we considered these explanations not reliable. Attribution maps and baseline values generated on the CIFAR-10 and the MNIST datasets are provided in Appendix G.6. Compared to zero/mean/blurring baseline values, our baseline values were more likely to ignore noisy variables in the background, which were far from the foreground in images. Compared to SHAP, our method yielded more informative attributions. Besides, our method generated smoother attributions than SAGE.

## 6 CONCLUSIONS

In this paper, we have defined the absence state of input variables in terms of causality. Then, we have found that most existing masking methods cannot faithfully remove existing causal patterns without triggering new patterns. In this way, we have formulated optimal baseline values for the computation of Shapley values as those that remove most causal patterns. Then, we have proposed an approximate-yet-efficient method to learn optimal baseline values that represent the absence states of input variables. Experimental results have demonstrated the effectiveness of our method.

## ETHIC STATEMENT

This paper aims to examine the masking approach in previous explaining methods. We find that previous settings of the masking approach cannot faithfully represent the absence of input variables, thereby hurting the trustworthiness of the obtained explanations. Therefore, we propose a new method to learn optimal baseline values to represent the absence of input variables. In this way, the trustworthiness of explanations of the DNN is further boosted. There are no ethical issues with this paper.

## REPRODUCIBILITY STATEMENT

We have provided proofs for all theoretical results in Appendix E and Appendix H. We have also provided experimental details in Section 5 and Appendix G. Furthermore, we will release the code when the paper is accepted.

## ACKNOWLEDGEMENT

This work is partially supported by the National Nature Science Foundation of China (62276165), National Key R&D Program of China (2021ZD0111602), Shanghai Natural Science Foundation (21JC1403800,21ZR1434600), National Nature Science Foundation of China (U19B2043). This work is also partially supported by Huawei Technologies Inc.

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

Table 7: Accuracy of Shapley values computed on the pre-defined decision tree, which was based on the MNIST dataset.

| Baseline value | zero | mean | baseline in SHAP | ours |
|---|---|---|---|---|
| Accuracy | 92.99% | 92.93% | 93.37% | **96.84%** |

Table 8: The learned baseline values could recover original samples from adversarial examples.

| | $\|\boldsymbol{x}^{\mathrm{adv}} - \boldsymbol{x}\|_2$ | $\|\boldsymbol{b} - \boldsymbol{x}\|_2$ |
|---|---|---|
| MNIST on LeNet | 2.33 | **0.43** |
| MNIST on AlexNet | 2.53 | **1.15** |
| CIFAR-10 on ResNet-20 | 1.19 | **1.11** |

## A  RELATED WORKS

*No previous methods directly examined the faithfulness of the masking methods. Instead, we made a survey in a larger scope of attribution methods and other explainable AI studies, and put them in the appendix. Nevertheless, we will put this section back to the main paper if the paper is accepted.*

In the scope of explainable AI, many methods (Simonyan et al., 2014; Yosinski et al., 2015; Mordvintsev et al., 2015; Dosovitskiy and Brox, 2016; Zhou et al., 2015) have been proposed to explain the DNN. Among all methods, the estimation of attributions for each input variable represents a classical direction (Zhou et al., 2016; Selvaraju et al., 2017; Lundberg and Lee, 2017; Shrikumar et al., 2017). In this paper, we mainly focus on attributions based on Shapley values.

**Shapley values.** The Shapley value (Shapley, 1953) in game theory was widely considered as a fair distribution of the overall reward in a game to each player (Weber, 1988). (Sen et al., 1981) and (Grömping, 2007) used the Shapley value to attribute the correlation coefficient of a linear regression to input features. (Štrumbelj et al., 2009; Štrumbelj and Kononenko, 2014) used the Shapley value to attribute the prediction of a model to input features. (Bork et al., 2004) used the Shapley value to measure importances of protein interactions in large, complex biological interaction networks. (Keinan et al., 2004) employed the Shapley value to measure causal effects in neurophysical models. (Sundararajan et al., 2017) proposed Integrated Gradients based on the AumannShapley(Aumann and Shapley, 2015) cost-sharing technique. Besides above local explanations, (Covert et al., 2020b) focused on the global interpretability.

In order to compute the Shapley value in deep models efficiently, (Lundberg and Lee, 2017) proposed various approximations for Shapley valus in DNNs. (Lundberg et al., 2018) further computed the Shapley value on tree emsembles. (Aas et al., 2021) generalized the approximation method in (Lundberg and Lee, 2017) to the case when features were related to each other. (Ancona et al., 2019) further formulated a polynomial-time approximation of Shapley values for DNNs.

**Baseline values.** In terms of baseline values of Shapley values, most studies (Covert et al., 2020a; Merrick and Taly, 2020; Sundararajan and Najmi, 2020; Kumar et al., 2020) compared influences of baseline values on explanations, without providing any principles for setting baseline values. Shrikumar et al. (2017) proposed DeepLIFT to estimate attributions of input variables, and also mentioned the choice of baseline values. Besides, Agarwal and Nguyen (2021) and Frye et al. (2021) used generative models to alleviate the out-of-distribution problem caused by baseline values. Unlike previous studies, we rethink and formulate baseline values from the perspective of game-theoretic causality. We define the absent state of input variables, and propose a method to learn optimal baseline values based on the number of causal patterns.

## B  QUANTITATIVE EVALUATION OF ATTRIBUTIONS FOR IMAGE CLASSIFICATION

In order to quantitatively evaluate Shapley values computed by different baseline values on the MNIST dataset, we constructed an And-Or decision tree following (Harradon et al., 2018), whose structure directly provided the ground-truth Shapley value for each input variable. Then, we used different attribution methods to explain the decision tree. Table 7 shows that our method generated more accurate Shapley values than other baseline values.

We constructed a decision tree (Song et al., 2013) for each category in the MNIST dataset. Specifically, for each category (digit), we first computed the average image over all training samples in this category. Let $\bar{\boldsymbol{x}}^{(c)} \in \mathbb{R}^n$ denote the average image of the $c$-th category. Then, we built a decision tree by considering each pixel as an internal node. The splitting rule for the decision tree was designed as follows. Given an input $\boldsymbol{x}$ in the category $c$, the splitting criterion at the pixel (node) $x_i$ was

designed as $\left((\bar{x}_i^{(c)} > 0.5) \& (x_i > 0.5)\right)^2$. If $(\bar{x}_i^{(c)} > 0.5) \& (x_i > 0.5) = $ True, then the pixel value $x_i$ was added to the output; otherwise, $x_i$ was ignored. In this way, the output of the decision tree was $f(\boldsymbol{x}) = \sum_{i \in V} x_i$, where $V = \{i \in N | (\bar{x}_i^{(c)} > 0.5) \& (x_i > 0.5) = $ True$\}$ denote the set of all pixels that satisfied the above equation. For inference, the probability of $\boldsymbol{x}$ belonging to the category $c$ was $p(c|\boldsymbol{x}) = \text{sigmoid}(\gamma(f(\boldsymbol{x}) - \beta))$, where $\gamma = 40$ was a constant and $\beta \propto \sum_{i \in N} \mathbb{1}_{\bar{x}_i^{(c)} > 0.5}$. In this case, we defined $v(\boldsymbol{x}_N) = \log \frac{p(c|\boldsymbol{x})}{1 - p(c|\boldsymbol{x})}$. Thus, the co-appearing of pixels in $V$ formed a causal pattern to contribute for $v(\boldsymbol{x}_N)$. In other words, because $\forall i \in N, x_i \geq 0$, the absence of any pixel in $V$ might deactivate this pattern by leading to a small probability $p(c|\boldsymbol{x}) < 0.5$ and a small $v$. This pattern can also be understood as an AND node in the And-Or decision tree (Song et al., 2013).

In the above decision tree, the ground-truth Shapley values of input variables (pixels) were easy to determine. The above decision tree ensured that the absence of any variable in $V$ would deactivate the causal pattern. Therefore, according to Theorem 2 in the paper, the output probability should be fairly assigned to pixels in $V$, *i.e.*, they shared the same Shapley values $\hat{\phi}_i = \frac{v(\boldsymbol{x}_N)}{|V|}$. For other pixels that were not contained in the output, their ground-truth Shapley values were zero.

We estimated Shapley values of input variables in the above decision tree by using zero baseline values, mean baseline values, baseline values in SHAP, and the learned baseline values by our method, respectively. Let $\phi_i$ denote the estimated Shapley value of the variable $i$. If $|\phi_i - \hat{\phi}_i| \leq 0.01$, we considered the estimated Shapley value $\phi_i$ correct; otherwise, incorrect. In this way, we computed the accuracy of the estimated Shapley values, and Table 7 shows that our method achieved the highest accuracy.

## C    REMOVING ADVERSARIAL PERTURBATIONS FROM THE INPUT

Let $\boldsymbol{x}$ denote the normal sample, and let $\boldsymbol{x}^{\text{adv}} = \boldsymbol{x} + \delta$ denote the adversarial example generated by (Madry et al., 2018). According to (Ren et al., 2021), the adversarial example $\boldsymbol{x}^{\text{adv}}$ mainly created out-of-distribution bivariate interactions with high-order contexts, which were actually related to the high-order interactions (causal patterns) in this paper. Thus, in the scenario of this study, the adversarial utility was owing to out-of-distribution high-order interactions (causal patterns). The removal of input variables was supposed to remove most high-order causal patterns.

Therefore, the baseline value can be considered as the recovery of the original sample. In this way, we used the adversarial example $\boldsymbol{x}^{\text{adv}}$ to initialize baseline values before learning, and used $L_{\text{marginal}}$ to learn baseline values. If the learned baseline values $\boldsymbol{b}$ satisfy $\|\boldsymbol{b} - \boldsymbol{x}\|_1 \leq \|\boldsymbol{x}^{\text{adv}} - \boldsymbol{x}\|_1$, we considered that our method successfully recovered the original sample to some extent. We conducted experiments using LeNet, AlexNet (Krizhevsky et al., 2012), and ResNet-20 on the MNIST dataset ($\|\delta\|_\infty \leq 32/255$) and the CIFAR-10 dataset ($\|\delta\|_\infty \leq 8/255$). Table 8 shows that our method recovered original samples from adversarial examples, which demonstrated the effectiveness of our method.

## D    AXIOMS OF THE SHAPLEY VALUE

The Shapley value (Shapley, 1953) was first introduced in game theory, which measures the contribution of each player in a game. Actually, given an input $\boldsymbol{x}$ with $n$ input variables, *i.e.*, $\boldsymbol{x} = [x_1, \ldots, x_n]$, we can consider a deep model as a game with $n$ players $N = \{1, 2, \cdots, n\}$. Each player $i$ is an input variable $x_i$ (*e.g.* an input dimension, a pixel, or a word). In this way, the problem of *fairly estimating attributions of input variables in the DNN* is equivalent to the problem of *fairly assigning the total reward in the game to each player*. The Shapley value is widely considered a fair attribution method, because it satisfies the following four axioms (Weber, 1988).
(1) *Linearity axiom*: If two games can be merged into a new game $u(\boldsymbol{x}_S) = v(\boldsymbol{x}_S) + w(\boldsymbol{x}_S)$, then Shapley values in the two old games also can be merged, *i.e.* $\forall i \in N$, $\phi_{i,u} = \phi_{i,v} + \phi_{i,w}$.
(2) *Dummy axiom and nullity axiom*: The dummy player $i$ is defined as a player without any interactions with other players, *i.e.* satisfying $\forall S \subseteq N \setminus \{i\}$, $v(\boldsymbol{x}_{S \cup \{i\}}) = v(\boldsymbol{x}_S) + v(\boldsymbol{x}_{\{i\}})$. Then, the dummy player's Shapley value is computed as $\phi_i = v(\boldsymbol{x}_{\{i\}})$. The null player $i$ is defined as a player that satisfies $\forall S \subseteq N \setminus \{i\}$, $v(\boldsymbol{x}_{S \cup \{i\}}) = v(\boldsymbol{x}_S)$. Then, the null player's Shapley value is $\phi_i = 0$.

---

[2]For Table 7, the splitting criterion was designed as $(\bar{x}_i^{(c)} > 0.5)$.

(3) *Symmetry axiom*: If $\forall S \subseteq N \setminus \{i,j\}$, $v(\boldsymbol{x}_{S\cup\{i\}}) = v(\boldsymbol{x}_{S\cup\{j\}})$, then $\phi_i = \phi_j$.
(4) *Efficiency axiom*: The overall reward of the game is equal to the sum of Shapley values of all players, *i.e.* $v(\boldsymbol{x}_N) - v(\boldsymbol{x}_\emptyset) = \sum_{i\in N} \phi_i$.

# E  PROOFS OF THEOREMS

This section provides proofs of theorems in the main paper.

## E.1  PROOF OF THEOREM 1

**Theorem 1** *(**Faithfulness**, proven by Ren et al. (2023a)) Let us consider a DNN $v$ and an input sample $\boldsymbol{x}$ with $n$ input variables. We can generate $2^n$ different masked samples, i.e., $\{\boldsymbol{x}_S | S \subseteq N\}$. The DNN's outputs on all masked samples can always be well mimicked as the sum of the triggered interaction effects in Eq. (3), i.e., $\forall S \subseteq N, v(\boldsymbol{x}_S) = \sum_{S'\subseteq S} U_{S'}$.*

***Proof:*** According to the definition of the Harsanyi dividend, we have $\forall S \subseteq \mathcal{N}$,

$$\sum_{S'\subseteq S} U_{S'} = \sum_{S'\subseteq S}\sum_{L\subseteq S'}(-1)^{|S'|-|L|}v(\boldsymbol{x}_L)$$

$$= \sum_{L\subseteq S}\sum_{S'\subseteq S:S'\supseteq L}(-1)^{|S'|-|L|}v(\boldsymbol{x}_L)$$

$$= \sum_{L\subseteq S}\sum_{s'=|L|}^{|S|}\sum_{\substack{S'\subseteq S:S\supseteq L \\ |S'|=s'}}(-1)^{s'-|L|}v(\boldsymbol{x}_L)$$

$$= \sum_{L\subseteq S}v(\boldsymbol{x}_L)\sum_{m=0}^{|S|-|L|}\binom{|S|-|L|}{m}(-1)^m = v(\boldsymbol{x}_S)$$

## E.2  PROOF OF THEOREM 2

**Theorem 2** *Harsanyi dividends can be considered as causal patterns of the Shapley value.*

$$\phi_i = \sum_{S\subseteq N\setminus\{i\}}\frac{1}{|S|+1}U_{S\cup\{i\}} \tag{8}$$

*In this way, the effect of an causal pattern consisting of $m$ variables can be fairly assigned to the $m$ variables. This connection has been proved in (Harsanyi, 1982).*

• *Proof:*

$$\text{right} = \sum_{S\subseteq N\setminus\{i\}}\frac{1}{|S|+1}U_{S\cup\{i\}}$$

$$= \sum_{S\subseteq N\setminus\{i\}}\frac{1}{|S|+1}\left[\sum_{L\subseteq S}(-1)^{|S|+1-|L|}v(L) + \sum_{L\subseteq S}(-1)^{|S|-|L|}v(L\cup\{i\})\right]$$

$$= \sum_{S\subseteq N\setminus\{i\}}\frac{1}{|S|+1}\sum_{L\subseteq S}(-1)^{|S|-|L|}\left[v(L\cup\{i\}) - v(L)\right]$$

$$= \sum_{L\subseteq N\setminus\{i\}}\sum_{K\subseteq N\setminus L\setminus\{i\}}\frac{(-1)^{|K|}}{|K|+|L|+1}\left[v(L\cup\{i\}) - v(L)\right] \qquad \% \text{ Let } K = S\setminus L$$

$$= \sum_{L\subseteq N\setminus\{i\}}\left(\sum_{k=0}^{n-1-|L|}\frac{(-1)^k}{k+|L|+1}\binom{n-1-|L|}{k}\right)\left[v(L\cup\{i\}) - v(L)\right] \qquad \% \text{ Let } k = |K|$$

$$= \sum_{L\subseteq N\setminus\{i\}}\frac{|L|!(n-1-|L|)!}{n!}\left[v(L\cup\{i\}) - v(L)\right] \qquad \% \text{ by the property of combinitorial number}$$

$$= \phi_i = \text{left}$$

Table 9: Comparison between ground-truth baseline values and incorrect baseline values. The last column shows ratios of causal patterns of different orders $r_m = \frac{\sum_{S \subseteq N, |S|=m} |U_S|}{\sum_{S \subseteq N, S \neq \emptyset} |U_S|}$. We consider interactions of input samples that activate causal patterns. We find that when models/functions contain a single complex collaborations between multiple variables (*i.e.* high-order causal patterns), incorrect baseline values usually generate a mixture of many low-order causal patterns. In comparison, ground-truth baseline values lead to sparse and high-order causal patterns.

| Functions ($\forall i \in N, i \in \{0,1\}$) | Baseline values $\boldsymbol{b}$ | Ratios $\boldsymbol{r}$ |
|---|---|---|
| $f(\boldsymbol{x}) = x_1 x_2 x_3 x_4 x_5$ 
 $\boldsymbol{x} = [1,1,1,1,1]$ | ground truth: $\boldsymbol{b}^* = [0,0,0,0,0]$ 
 incorrect: $\boldsymbol{b}^{(1)} = [0.5,0.5,0.5,0.5,0.5]$ 
 incorrect: $\boldsymbol{b}^{(2)} = [0.1,0.2,0.6,0.0,0.1]$ 
 incorrect: $\boldsymbol{b}^{(3)} = [0.7,0.1,0.3,0.5,0.1]$ |  |
| $f(\boldsymbol{x}) = sigmoid(5x_1 x_2 x_3 + 5x_4 - 7.5)$ 
 $\boldsymbol{x} = [1,1,1,1]$ | ground truth: $\boldsymbol{b}^* = [0,0,0,0]$ 
 incorrect: $\boldsymbol{b}^{(1)} = [0.5,0.5,0.5,0.5]$ 
 incorrect: $\boldsymbol{b}^{(2)} = [0.6,0.4,0.7,0.3]$ 
 incorrect: $\boldsymbol{b}^{(3)} = [0.3,0.6,0.5,0.8]$ |  |
| $f(\boldsymbol{x}) = x_1(x_2 + x_3 - x_4)^3$ 
 $\boldsymbol{x} = [1,1,1,0]$ | ground truth: $\boldsymbol{b}^* = [0,0,0,1]$ 
 incorrect: $\boldsymbol{b}^{(1)} = [0.5,0.5,0.5,0.5]$ 
 incorrect: $\boldsymbol{b}^{(2)} = [0.2,0.3,0.6,0.1]$ 
 incorrect: $\boldsymbol{b}^{(3)} = [1.0,0.3,1.0,0.1]$ |  |

## E.3 PROOF OF REMARK2, THEOREM 3, AND THEOREM 4

**Remark 2** *Let us consider a function with a single causal pattern $f(\boldsymbol{x}_S) = w_S \prod_{j \in S}(x_j - \delta_j)$. Accordingly, ground-truth baseline values of variables are obviously $\{\delta_j\}$, because setting any variable $\forall j \in S, x_j = \delta_j$ will deactivate this pattern. Given the correct baseline values $b_j^* = \delta_j$, we can use a single causal pattern to regress $f(\boldsymbol{x}_S)$, i.e., $U_S = f(\boldsymbol{x}_S), \forall S' \neq S, U_{S'} = 0$.*

**Theorem 3** *For the function $f(\boldsymbol{x}_S) = w_S \prod_{j \in S}(x_j - \delta_j)$, if we use $m'$ incorrect baseline values $\{b_j' | b_j' \neq \delta_j\}$ to replace correct ones to compute causal effects, then the function will be explained to contain at most $2^{m'}$ causal patterns.*

**Theorem 4** *If we use $m'$ incorrect baseline values to compute causal effects in the function $f(\boldsymbol{x}_S) = w_S \prod_{j \in S}(x_j - \delta_j)$, a total of $\binom{m'}{k - |S| + m'}$ causal patterns of the $k$-th order emerge, $k \geq |S| - m'$. A causal pattern of the $k$-th order means that this causal pattern represents the AND relationship between $k$ variables.*

• **Theoretical proof:** Without loss of generality, let us consider an input sample $\boldsymbol{x}$, with $\forall j \in S, x_j \neq \delta_j$. Based on the ground-truth baseline value $\{\delta_j\}$, we have

(1) $v(\boldsymbol{x}_S) = f(\boldsymbol{x}_S) = w_S \prod_{j \in S}(x_j - \delta_j) \neq 0$,

(2) $\forall S' \subsetneq S, v(\boldsymbol{x}_{S'}) = w_S \prod_{j \in S'}(x_j - \delta_j) \prod_{k \in S \setminus S'}(\delta_k - \delta_k) = 0$,
Accordingly, we have $U_S = \sum_{S' \subseteq S}(-1)^{|S|-|S'|} v(\boldsymbol{x}_{S'}) = v(\boldsymbol{x}_S) \neq 0$. For $S' \subsetneq S$, we have $U_{S'} = \sum_{L \subseteq S'}(-1)^{|S'|-|L|} v(\boldsymbol{x}_L) = \sum_{L \subseteq S'} 0 = 0$.

(3) $\forall S' \neq S$, let $S' = L \cup M$, where $L \subseteq S$ and $M \cap S = \emptyset$. Then, we have

$$U_{S'} = \sum_{T \subseteq S'} (-1)^{|S'|-|T|} v(\boldsymbol{x}_T)$$

$$= \sum_{\substack{L' \subseteq L \\ L' \neq \emptyset}} (-1)^{|S'|-|L'|} v(\boldsymbol{x}_{L'}) + \sum_{\substack{M' \subseteq M \\ M' \neq \emptyset}} (-1)^{|S'|-|M'|} \underbrace{v(\boldsymbol{x}_{M'})}_{=v(\boldsymbol{x}_\emptyset)=0}$$

$$+ \sum_{\substack{L' \subseteq L, M' \subseteq M \\ L' \neq \emptyset, M' \neq \emptyset}} (-1)^{|S|-|L'|-|M'|} \underbrace{v(\boldsymbol{x}_{L' \cup M'})}_{=v(L')} + (-1)^{|S'|} \underbrace{v(\boldsymbol{x}_\emptyset)}_{=0}$$

$$= \sum_{\substack{L' \subseteq L \\ L' \neq \emptyset}} (-1)^{|S'|-|L'|} v(\boldsymbol{x}_{L'}) + \sum_{\substack{L' \subseteq L, M' \subseteq M \\ L' \neq \emptyset, M' \neq \emptyset}} (-1)^{|S|-|L'|-|M'|} v(\boldsymbol{x}_{L'})$$

$$= (-1)^{|S'|-|S|} v(\boldsymbol{x}_S) + \sum_{\substack{M' \subseteq M \\ M' \neq \emptyset}} (-1)^{|S'|-|S|-|M'|} v(\boldsymbol{x}_S) \quad \% \ v(\boldsymbol{x}_{L'}) \neq 0 \text{ only if } L' = S$$

$$= \sum_{M' \subseteq M} (-1)^{|S'|-|S|-|M'|} v(\boldsymbol{x}_S) = 0$$

Therefore, there is only one causal pattern with non-zero effect $U_S$.

In comparison, if we use $m'$ incorrect baseline values $\{\delta'_j\}$, where $\sum_{j \in S} \mathbb{1}_{\delta'_j \neq \delta_j} = m'$, then the function will be explained to contain at most $2^{m'}$ causal patterns. For the simplicity of notations, let $S = \{1, 2, ..., m\}$, and $\delta'_1 = \delta_1 + \epsilon_1, ..., \delta'_{m'} = \delta_{m'} + \epsilon_{m'}$, where $\epsilon_1, ..., \epsilon_{m'} \neq 0$. Let $T = \{1, 2, \ldots, m'\}$. In this case, we have
(1) $v(\boldsymbol{x}_S) = f(\boldsymbol{x}_S) \neq 0$
(2) $\forall S' \subsetneq S, |S'| < m - m', v(\boldsymbol{x}_{S'}) = w_S \prod_{j \in S'} (x_j - \delta_j) \prod_{l \in S \setminus S'} (\delta'_l - \delta_l)$. Because $|S| - |S'| > m'$, there is at least one variable with ground-truth baseline value in $S \setminus S'$. Therefore, $v(\boldsymbol{x}_{S'}) = 0$. Furthermore, $U_{S'} = \sum_{L \subseteq S'} (-1)^{|S'|-|L|} v(\boldsymbol{x}_L) = 0$
(3) $\forall S' \subsetneq S, |S'| = k \geq m - m', v(\boldsymbol{x}_{S'}) = w_S \prod_{j \in S'} (x_j - \delta_j) \prod_{l \in S \setminus S'} (\delta'_l - \delta_l)$. If $S \setminus T \subseteq S'$, then $S \setminus S' \subseteq T$ and $v(\boldsymbol{x}_{S'}) \neq 0$. Otherwise, $v(\boldsymbol{x}_{S'}) = 0$. Then,

$$U_{S'} = \sum_{L \subseteq S'} (-1)^{|S'|-|L|} v(\boldsymbol{x}_L)$$

$$= \sum_{L \subseteq S', |L| < m-m'} (-1)^{|S'|-|L|} v(\boldsymbol{x}_L) + \sum_{L \subseteq S', L \geq m-m'} (-1)^{|S'|-|L|} v(\boldsymbol{x}_L)$$

$$= 0 + \sum_{L \subseteq S', L \geq m-m', L \supseteq S \setminus T} (-1)^{|S'|-|L|} v(\boldsymbol{x}_L) + \sum_{L \subseteq S', L \geq m-m', L \not\supseteq S \setminus T} (-1)^{|S'|-|L|} v(\boldsymbol{x}_L)$$

$$= \sum_{L \subseteq S', L \geq m-m', L \supseteq S \setminus T} (-1)^{|S'|-|L|} v(\boldsymbol{x}_L)$$

If the above $U_{S'} = 0$, it indicates that $S \setminus T \not\subseteq S'$. In this case, there is no subset $L \subseteq S'$ s.t. $S \setminus T \subseteq L$. In other words, only if $S \setminus T \subseteq S', U_{S'} \neq 0$. In this way, a total of $\binom{m'}{k-(|S|-m')}$ causal patterns of the $k$-th order emerge, where the order $k$ of a causal pattern means that this causal pattern $S'$ contains $k = |S'|$ variables. There are totally $\sum_{k=|S|-m'}^{m} \binom{m'}{k-(|S|-m')} = 2^{m'}$ causal patterns in $\boldsymbol{x}$.

For example, if the input $\boldsymbol{x}$ is given as follows,

$$x_i = \begin{cases} \delta_i + 2\epsilon_i, & i \in T = \{1, \ldots, m'\} \\ \delta_i + \epsilon_i, & i \in S \setminus T = \{m' + 1, \ldots, m\} \end{cases}$$

where $\epsilon_i \neq 0$ are arbitrary non-zero scalars. In this case, we have $\forall S' \subseteq T, U_{S' \cup \{m'+1,\ldots,m\}} = \epsilon_1 \epsilon_2 ... \epsilon_m \neq 0$. Besides, if $\{m' + 1, ..., m\} \not\subseteq S'$, we have $U_{S'} = 0$. In this way, there are totally $2^{m'}$ causal patterns in $\boldsymbol{x}$.

• *Experimental verification:* We further conducted experiments to show that the incorrect setting of baseline values makes a model/function consisting of high-order causal patterns be mistakenly explained as a mixture of low-order and high-order causal patterns. To show this phenomenon, we compare causal patterns computed using ground-truth baseline values and incorrect baseline values

in Table 9, and the results verify our conclusion. We find that when models/functions contain complex collaborations between multiple variables (*i.e.* high-order causal patterns), incorrect baseline values usually generate fewer high-order causal patterns and more low-order causal patterns than ground-truth baseline values. In other words, the model/function is explained as massive low-order causal patterns. In comparison, ground-truth baseline values lead to sparse and high-order salient patterns.

## F    PROVING THAT MASKING INPUT VARIABLES REMOVES CAUSAL EFFECTS

In this section, we prove that for the causal pattern $S \ni i$, if the input variable $i$ is masked, then the causal effect $w_S = 0$.

***Proof:*** let $S = S' \cup \{i\}$. If $i \in S$ is masked, then $\forall L \ s.t. \ i \notin L, \boldsymbol{x}_L = \boldsymbol{x}_{L \cup \{i\}}$. Therefore, $v(L \cup \{i\}) = v(L)$. According to the definition of Harsanyi dividend (Harsanyi, 1982), we have

$$
\begin{aligned}
U_S &= \sum_{L \subseteq S} (-1)^{|S|-|L|} v(L) \\
&= \sum_{L \subseteq (S' \cup \{i\})} (-1)^{|S'|+1-|L|} v(L) \\
&= \sum_{L \subseteq S'} (-1)^{|S'|+1-|L|} v(L) + \sum_{L \subseteq S'} (-1)^{|S'|-|L|} v(L \cup \{i\}) \\
&= \sum_{L \subseteq S'} (-1)^{|S'|+1-|L|} v(L) + \sum_{L \subseteq S'} (-1)^{|S'|-|L|} v(L) \\
&= \sum_{L \subseteq S'} \left( (-1)^{|S'|+1-|L|} + (-1)^{|S'|-|L|} \right) v(L) \\
&= \sum_{L \subseteq S'} (-1+1)(-1)^{|S'|-|L|} v(L) \\
&= 0
\end{aligned}
$$

Note that the causal pattern not containing $i$ will not be deactivated by the masking of $i$. For example, $\{eyes, beak\}$ is not deactivated by the absence of *forehead*, because this pattern represents the AND relationship between *eyes* and *beak*, and it does not contain *forehead*.

## G    MORE EXPERIMENTAL DETAILS AND RESULTS

### G.1    VERIFICATION OF THE SPARSITY OF CAUSAL PATTERNS

In this subsection, we conducted experiments to verify the sparsity of causal effects, which is introduced in Remark 1. To this end, we computed causal effects $U_S$ of all $2^n$ causal patterns encoded by a DNN. Specifically, we trained a three-layer MLP on the income dataset and computed causal effects in the model. Figure 4 shows the distribution of absolute causal effects $|U_S|$ of causal patterns in the first five samples of each category of the income dataset. These results show that most causal patterns had insignificant causal effects, $U_S \approx 0$. Only a few causal patterns had salient causal effects.

Moreover, we also conducted experiments to demonstrate the universality of this phenomenon. We trained the five-layer MLP, CNN, LSTM, ResNet-32, and VGG-16 on the UCI census income dataset, the UCI TV news channel commercial detection dataset, the SST-2 dataset, and the MNIST dataset, respectively. Figure 5 shows the absolute causal effects $U_S$ in the descending order. These results show that various DNNs learned on different tasks could be explained by a set of sparse causal patterns.

### G.2    VERIFICATION OF USING CAUSAL PATTERNS TO EXAMINE THE STATE OF INPUT VARIABLES

In this subsection, we conducted experiments to verify that causal patterns reflect the states of removing existing patterns. Given causal effects $U_S$ in the normal input image and causal effects

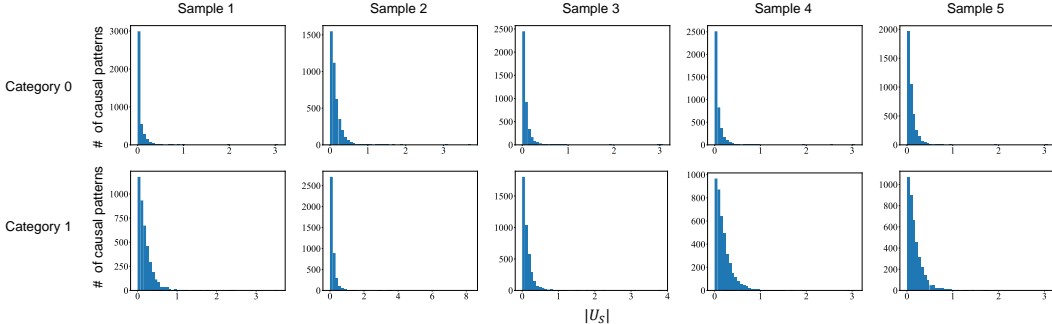

Figure 4: Histograms of absolute causal effects of causal patterns encoded by the three-layer MLP trained on the income dataset.

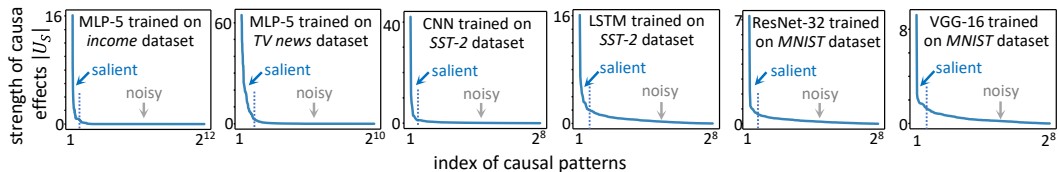

Figure 5: Absolute causal effects of different causal patterns shown in a descending order, which shows that sparse causality is universal for various DNNs.

$U_S^{(\text{noise})}$ in the white noise input, we compared their distributions in Figure 6. Note that we assumed that the white noise input naturally contained information for classification than the normal input image. We found that most causal effects in the white noise input were close to zero, and there were few salient causal patterns. Besides, we computed the average strength of causal effects in the above two inputs. In the normal input, the average strength of causal effects $\mathbb{E}_{S \subseteq N}|U_S| = 5.5285$, while in the white noise input, the average strength was much smaller, $\mathbb{E}_{S \subseteq N}|U_S^{(\text{noise})}| = 0.2321$. These results indicated that salient causal patterns could reflect the information encoded in the input.

Figure 6: The distribution of causal effects $U_S$ in the normal input, and the distribution of causal effects $U_S^{(\text{noise})}$ in the white noise input.

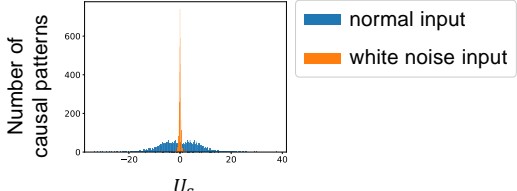

### G.3 EFFECTS OF THE PROPOSED METHOD ON MULTI-ORDER SHAPLEY VALUES AND MULTI-ORDER MARGINAL BENEFITS

In this section, we conducted experiments to verify that baseline values $\boldsymbol{b}$ learned by the proposed loss function in Eq. (7) could effectively reduce causal effects of low-order causal patterns in Eq. (5).

To this end, we computed the metric $\mathbb{E}_{\boldsymbol{x}}[(\mathbb{E}_{|S|=m}|U_S|)/(v(\boldsymbol{x}_N) - v(\boldsymbol{x}_\emptyset))]$ to measure the relative strength of causal patterns of a specific order $m$, in order to evaluate the effectiveness of baseline values. Fig. 7(a) shows that compared to zero baseline values, **our method effectively reduced low-order causal patterns.** In addition, Fig. 8 and Fig. 7(b) verify that the loss $L_{\text{Shapley}}$ in Eq. (7) **reduced the number of salient causal patterns in** $\Omega$, which means $L_{\text{Shapley}}$ avoided the exponential number of causal patterns caused by incorrect baseline values.

### G.4 DISCUSSION ABOUT THE SETTING OF GROUND-TRUTH BASELINE VALUES.

This section discusses the ground truth of baseline values of synthetic functions in Section 5.1 of the main paper. In order to verify the correctness of the learned baseline values, we conducted experiments on synthetic functions with ground-truth baseline values. We randomly generated 100

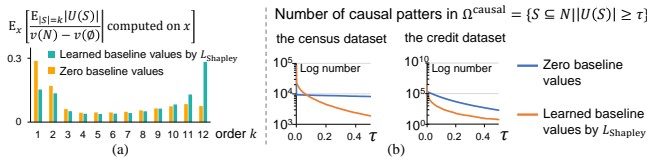
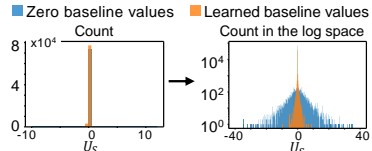

Figure 7: Our method successfully (a) reduced effects of low-order causal patterns on the MLP trained on the census dataset, and (b) reduced the number of salient patterns.

Figure 8: Distribution of causal effects $U_S$ of causal patterns in 20 samples in the credit dataset.

Table 10: Examples of synthetic functions and their ground-truth baseline values.

| Functions ($\forall i \in N, x_i \in \{0,1\}$) | The ground truth of baseline values |
|---|---|
| $-0.185x_1(x_2 + x_3)^{2.432} - x_4x_5x_6x_7x_8x_9x_{10}x_{11}$ | $b_i^* = 0$ for $i \in \{1, \ldots, 11\}$ |
| $-sigmoid(-4x_1 - 4x_2 - 4x_3 + 2) - 0.011x_4(x_5 + x_6 + x_7 + x_8 + x_9 + x_{10} + x_{11})^{2.341}$ | $b_i^* = 1$ for $i \in \{1, 2, 3\}, b_i^* = 0$ for $i \in \{4, \ldots, 11\}$ |
| $0.172x_1x_2x_3(x_4 + x_5)^{2.543} - 0.171x_6(x_7 + x_8)^{2.545} - 0.093(x_9 + x_{10} + x_{11})^{2.157}$ | $b_i^* = 0$ for $i \in \{1, \ldots, 11\}$ |
| $-sigmoid(5x_1 + 5x_2x_3x_4 - 7.5) - sigmoid(-8x_5x_6 + 8x_7 + 4) + x_8x_9x_{10}x_{11}$ | $b_i^* = 1$ for $i = 7, b_i^* = 0$ for $i \in \{1, 2, 3, 4, 5, 6, 8, 9, 10, 11\}$ |
| $-x_1x_2x_3 + 0.156(x_4 + x_5 + x_6)^{1.693} - x_7x_8x_9x_{10}$ | $b_i^* = 0$ for $i \in \{1, \ldots, 10\}$ |

functions whose causal patterns and ground truth of baseline values could be easily determined. As Table 10 shows, the generated functions were composed of addition, subtraction, multiplication, exponentiation, and *sigmoid* operations.

The ground truth of baseline values in these functions was determined based on causal patterns between input variables. In order to represent the absence states of variables, baseline values should activate as few salient patterns as possible, where activation states of causal patterns were considered as the most infrequent state. Thus, we first identified the activation states of causal patterns of variables, and the ground-truth of baseline values was set as values that inactivated causal patterns under different masks. We took the following examples to discuss the setting of ground-truth baseline values (in the following examples, $\forall i \in N, x_i \in \{0, 1\}$ and $b_i^* \in \{0, 1\}$).

• $f(\boldsymbol{x}) = x_1x_2x_3 + sigmoid(x_4 + x_5 - 0.5) \cdots$. Let us just focus on the term of $x_1x_2x_3$ in $f(\boldsymbol{x})$. The activation state of this causal pattern is $x_1x_2x_3 = 1$ when $\forall i \in \{1, 2, 3\}, x_i = 1$. In order to inactivate the causal pattern, we set $\forall i \in \{1, 2, 3\}, b_i^* = 0$.

• $f(\boldsymbol{x}) = -x_1x_2x_3 + (x_4 + x_5)^3 + \cdots$. Let us just focus on the term of $-x_1x_2x_3$ in $f(\boldsymbol{x})$. The activation state of this causal pattern is $-x_1x_2x_3 = -1$ when $\forall i \in \{1, 2, 3\}, x_i = 1$. In order to inactivate the causal pattern, we set $\forall i \in \{1, 2, 3\}, b_i^* = 0$.

• $f(\boldsymbol{x}) = (x_1 + x_2 - x_3)^3 + \cdots$. Let us just focus on the term of $(x_1 + x_2 - x_3)^3$ in $f(\boldsymbol{x})$. The activation state of this causal pattern is $(x_1 + x_2 - x_3)^3 = 8$ when $x_1 = x_2 = 1, x_3 = 0$. In order to inactivate the causal pattern under different masks, we set $b_1^* = b_2^* = 0, b_3^* = 1$.

• $f(\boldsymbol{x}) = sigmoid(3x_1x_2 - 3x_3 - 1.5) + \cdots$. Let us just focus on the term of $sigmoid(3x_1x_2 - 3x_3 - 1.5)$ in $f(\boldsymbol{x})$. In this case, $x_1, x_2, x_3$ form a salient causal pattern because $sigmoid(3x_1x_2 - 3x_3 - 1.5) > 0.5$ only if $x_1 = x_2 = 1$ and $x_3 = 0$. Thus, in order to inactivate causal patterns, ground-truth baseline values are set to $b_1^* = b_2^* = 0, b_3^* = 1$.

**Ground-truth baseline values of functions in (Tsang et al., 2018).** This section provides more details about ground-truth baseline values of functions proposed in (Tsang et al., 2018). We evaluated the correctness of the learned baseline values using functions proposed in (Tsang et al., 2018). Among all the 92 input variables in these functions, the ground truth of 61 variables could be determined and are reported in Table 11. Note that some variables cannot be 0 or 1 (*e.g.* $x_8$ cannot be zero in the first function), and we set $\forall i \in N, x_i \in \{0.001, 0.999\}$ for variables in these functions instead. Similarly, we set the ground truth of baseline values $\forall i \in N, b_i^* \in \{0.001, 0.999\}$. Some variables did not collaborate/interact with other variables (*e.g.* $x_4$ in the first function), thereby having no causal patterns. We did not assign ground-truth baseline values for these individual variables, and these variables are not used for evaluation. Some variables formed more than one causal pattern with other variables, and had different ground-truth baseline values *w.r.t.* different patterns. In this case, the collaboration between input variables was complex and hard to analyze, so we did not consider such input variables with conflicting patterns for evaluation, either.

Table 11: Functions in (Tsang et al., 2018) and their ground-truth baseline values.

| Functions ($\forall i \in N, x_i \in \{0.001, 0.999\}$) | The ground truth of baseline values |
|---|---|
| $\pi^{x_1 x_2}\sqrt{2x_3} - \sin^{-1}(x_4) + \log(x_3 + x_5) - \frac{x_9}{x_{10}}\sqrt{\frac{x_7}{x_8}} - x_2 x_7$ | $b_i^* = 0.999$ for $i \in \{5, 8, 10\}, b_i^* = 0.001$ for $i \in \{1, 2, 7, 9\}$ |
| $\pi^{x_1 x_2}\sqrt{2|x_3|} - \sin^{-1}(0.5x_4) + \log(|x_3 + x_5| + 1) + \frac{x_9}{1+|x_{10}|}\sqrt{\frac{x_7}{1+|x_8|}} - x_2 x_7$ | $b_i^* = 0.999$ for $i = 5, b_i^* = 0.001$ for $i \in \{1, 2, 7, 9\}$ |
| $\exp|x_1 - x_2| + |x_2 x_3| - x_3^{2|x_4|} + \log(x_4^2 + x_5^2 + x_7^2 + x_8^2) + x_9 + \frac{1}{1+x_{10}^2}$ | $b_i^* = 0.999$ for $i \in \{3, 5, 7, 8\}$ |
| $\exp|x_1 - x_2| + |x_2 x_3| - x_3^{2|x_4|} + (x_1 x_4)^2 + \log(x_4^2 + x_5^2 + x_7^2 + x_8^2) + x_9 + \frac{1}{1+x_{10}^2}$ | $b_i^* = 0.999$ for $i \in \{3, 5, 7, 8\}$ |
| $\frac{1}{1+x_1^2+x_2^2+x_3^2} + \sqrt{\exp(x_4 + x_5)} + |x_6 + x_7| + x_8 x_9 x_{10}$ | $b_i^* = 0.999$ for $i \in \{1, 2, 3\}, b_i^* = 0.001$ for $i \in \{4, 5, 8, 9, 10\}$ |
| $\exp(|x_1 x_2| + 1) - \exp(|x_3 + x_4| + 1) + \cos(x_5 + x_6 - x_8) + \sqrt{x_8^2 + x_9^2 + x_{10}^2}$ | $b_i^* = 0.999$ for $i \in \{8, 9, 10\}, b_i^* = 0.001$ for $i \in \{1, 2, 3, 4, 5, 6\}$ |
| $(\arctan(x_1) + \arctan(x_2))^2 + \max(x_3 x_4 + x_6, 0) - \frac{1}{1+(x_4 x_5 x_6 x_7 x_8)^2} + \left(\frac{|x_7|}{1+|x_9|}\right)^5 + \sum_{i=1}^{10} x_i$ | $b_i^* = 0.999$ for $i = 9, b_i^* = 0.001$ for $i \in \{1, 2, 3, 4, 5, 6, 7, 8\}$ |
| $x_1 x_2 + 2^{x_3 + x_5 + x_6} + 2^{x_3 + x_4 + x_5 + x_7} + \sin(x_7 \sin(x_8 + x_9)) + \arccos(0.9x_{10})$ | $b_i^* = 0.001$ for $i \in \{1, 2, 3, 4, 5, 6\}$ |
| $\tanh(x_1 x_2 + x_3 x_4)\sqrt{|x_5|} + \exp(x_5 + x_6) + \log((x_6 x_7 x_8)^2 + 1) + x_9 x_{10} + \frac{1}{1+|x_{10}|}$ | $b_i^* = 0.001$ for $i \in \{6, 7, 8, 9, 10\}$ |
| $\sinh(x_1 + x_2) + \arccos(\tanh(x_3 + x_5 + x_7)) + \cos(x_4 + x_5) + \sec(x_7 x_9)$ | $b_i^* = 0.999$ for $i = 3, b_i^* = 0.001$ for $i \in \{1, 2, 4\}$ |

## G.5 DISCUSSION ABOUT THE SETTING OF GROUND-TRUTH SHAPLEY VALUES.

This section discusses the ground truth of Shapley values in the extended Addition-Multiplication dataset (Zhang et al., 2021c), which is used in Section 5.1 of the main paper. In order to verify the correctness of the Shapley values obtained by the optimal baseline values in this paper, we conducted experiments on the extended Addition-Multiplication dataset (Zhang et al., 2021c) with ground-truth Shapley values.

The Addition-Multiplication dataset in (Zhang et al., 2021c) contained functions that only consisted of addition and multiplication operations. For example, $f(\boldsymbol{x}) = x_1 x_2 + x_3 x_4$ where each input variable $x_i \in \{0, 1\}$ was a binary variable. Given $\boldsymbol{x} = [1, 1, 1, 1]$, the function contained two salient causal patterns, *i.e.* $\{x_1, x_2\}$ and $\{x_3, x_4\}$, and their benefits to the output were $U_{\{x_1, x_2\}} = U_{\{x_3, x_4\}} = 1$, respectively. According to (Harsanyi, 1982), the Shapley value is a uniform distribution of attributions. Therefore, the effect of a causal pattern was supposed to be uniformly assigned to variables in the pattern. Thus, the ground-truth Shapley values of variables were $\hat{\phi}_1 = \hat{\phi}_2 = 1/2$, and $\hat{\phi}_3 = \hat{\phi}_4 = 1/2$. However, if the input $\boldsymbol{x} = [1, 0, 1, 1]$, then the pattern $\{x_1, x_2\}$ was deactivated and $U_{\{x_1, x_2\}} = 0$. In this case, $\hat{\phi}_1 = \hat{\phi}_2 = 0$ while $\hat{\phi}_3 = \hat{\phi}_4 = 1/2$.

According to the analysis in Appendix G.4, ground-truth baseline values in the Addition-Multiplication dataset were all zero. Then our method is equivalent to the zero baseline values. Therefore, in order to avoid all ground-truth baseline values being zero, we added the subtraction operation. We also added a coefficient before each term in the function to boost the diversity of functions. For example, $f(\boldsymbol{x}) = 3.2x_1 x_2 + 1.5x_3(x_4 - 1)$. This function also contained two causal patterns, but the ground-truth baseline values of variables were different from the aforementioned function. Here, $b_0^* = b_0^* = b_3^* = 0$ and $b_4^* = 1$. Given the input $\boldsymbol{x} = [1, 1, 1, 0]$, all patterns were activated and $f(\boldsymbol{x}) = U_{\{x_1, x_2\}} + U_{\{x_3, x_4\}} = 3.2 + (-1.5) = 1.7$. Ground-truth Shapley values of input variables were $\hat{\phi}_1 = \hat{\phi}_2 = 3.2/2$ and $\hat{\phi}_3 = \hat{\phi}_4 = -1.5/2$. However, for the input $\boldsymbol{x} = [1, 1, 1, 1]$, the pattern $\{x_3, x_4\}$ was deactivated, thereby $\hat{\phi}_3 = \hat{\phi}_4 = 0$. Note that the above function can also be considered to contain three patterns ($f(\boldsymbol{x}) = 3.2x_1 x_2 + 1.5x_3 x_4 - 1.5x_3$). According to Occam's Razor, we follow the principle of the most simplified interaction to recognize causal patterns in the function, *i.e.* using the least number of causal patterns. Thus, we consider the above function $f(\boldsymbol{x}) = 3.2x_1 x_2 + 1.5x_3(x_4 - 1)$ containing two salient causal patterns.

Based on the extended Addition-Multiplication dataset, we randomly generated an input sample for each function in the dataset. Each variable $x_i$ in input samples were independently sampled following the Bernoulli distribution, *i.e.* $p(x_i = 1) = 0.7$. Therefore, for the mean baseline, baseline values of different input variables were all $0.7$. For the baseline value based on the marginal distribution, which was used in SHAP (Lundberg and Lee, 2017), $p(x_i') \sim Bernoulli(0.7)$. Then, we compared the accuracy of the computed Shapley values of input variables based on zero baseline values, mean baseline values, baseline values in SHAP, and the optimal baseline values defined in this paper, respectively. The result in Table 4 of the main paper shows that the optimal baseline values correctly generated the ground-truth attributions/Shapley values of input variables.

## G.6 EXPERIMENTAL RESULTS ON THE MNIST AND THE CIFAR-10 DATASETS

**Experimental results on the MNIST datasets.** Experimental settings on the MNIST dataset have been introduced in Section 5.2 of the paper. Figure 9 shows the learned baseline values on the

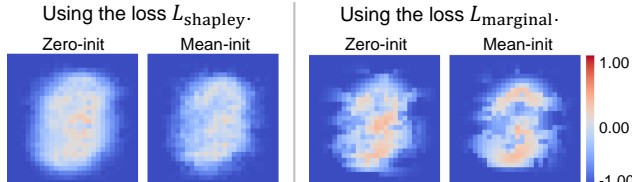

Figure 9: The learned baseline values on the MNIST dataset (better viewed in color).

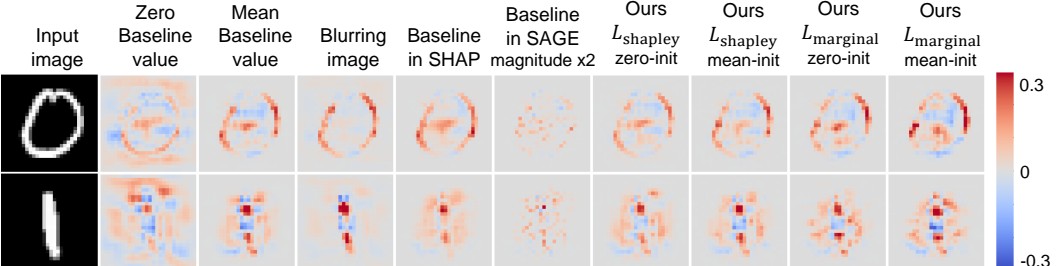

Figure 10: Shapley values computed with different baseline values on the MNIST dataset.

MNIST dataset by using $L_{\text{Shapley}}$ and $L_{\text{marginal}}$ in Section 4 of the paper, respectively. Figure 10 shows the computed Shapley values using different baseline values on the MNIST dataset. Compared to zero/mean/blurring baseline values, the learned baseline values by our method removed noisy variables on the background, which were far from digits in images. Compared to SHAP, our method yielded more informative attributions. Shapley values computed using baseline values in SAGE were dotted. In comparison, our method generated smoother attributions.

**Experimental results on the CIFAR-10 datasets.** First, let us clarify the experiment settings on the CIFAR-10 dataset. We split each image into $8 \times 8$ grids. Each grid contained $4 \times 4$ pixels, which shared a specific RGB color as their baseline value. Thus, we needed to learn a total of $8 \times 8 \times 3$ color values as baseline values. We learned baseline values by using the loss $L_{\text{Shapley}}$. Then, we computed Shapley values for grids by using zero baseline values and computed Shapley values by using the learned baseline values for comparison. Figure 11 shows that Shapley values computed with the learned baseline values mainly focused on objects in images. In comparison, Shapley values based on zero baselines values mainly focused on the background.

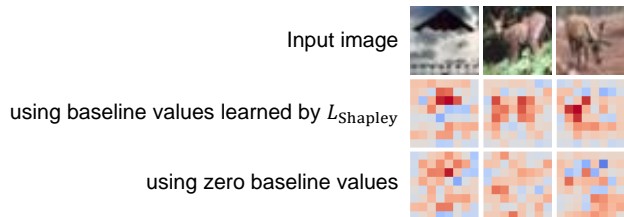

Figure 11: Shapley values computed with different baseline values. Images were selected from the CIFAR-10 dataset.

### G.7 EXPERIMENTAL RESULTS ON THE UCI SOUTH GERMAN CREDIT DATASET.

This section provides experimental results on the UCI South German Credit dataset (Dua and Graff, 2017). Based on the UCI datasets, we learned MLPs following settings in (Guidotti et al., 2018). Figure 12 compares Shapley values computed using different baseline values. Just like results on the UCI Census Income dataset, attributions (Shapley values) generated by our learned baseline values are similar to results of the varying baseline values in SHAP and SAGE. However, the zero/mean baseline values usually generated conflicting results with all other methods.

Besides, we also noticed that baseline values learned with different initialization settings (*zero-init* and *mean-init*) all converged to similar baseline values, except for very few dimensions having multiple local-minimum solutions, which proved the stability of our method. More specifically, an input variable might have different optimal baseline values in real applications from different

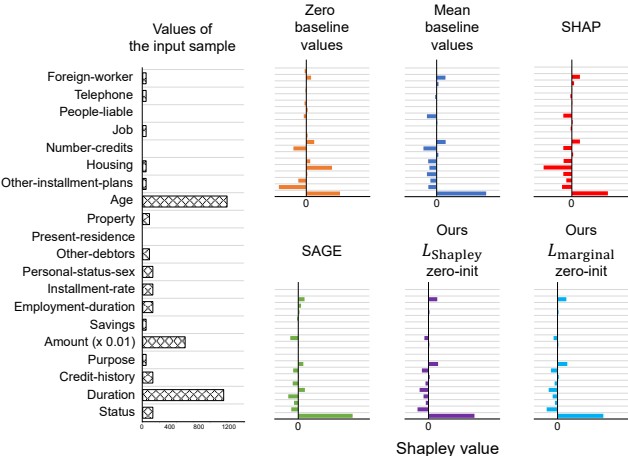

Figure 12: Shapley values computed with different baseline values on the UCI South German Credit dataset.

perspectives. For example, in the function of $y = (a-0.5)(b-3)(c+2)+(a+0.5)(d-3)(e+2)+f$, the variable $a$ had two equivalent baseline value $\pm 0.5$.

### G.8 Computational complexity of the proposed loss functions

In this section, we will introduce both the theoretical complexity of the loss function and the real complexity of approximating the loss in real applications.

In terms of the theoretical complexity, the complexity of the loss functions $L_{\text{Shapley}}$ and $L_{\text{marginal}}$ in Eq. (7) is $Dn \cdot \sum_{m=0}^{\lambda} \binom{n}{m}$, where $D$ denotes the number of samples $\boldsymbol{x}$ in the dataset. $n$ is the number of input variables in the input. $\sum_{m=0}^{\lambda} \binom{n}{m}$ denotes the computational complexity of all terms $\phi_i^{(m)}(x|b)$ of different orders $m \leq \lambda$.

In terms of real implementation, as mentioned in Section 5.2, we used the sampling-based approximation (Castro et al., 2009) to compute the loss function. We randomly sampled $K$ input variables in each epoch, and we randomly sampled $T$ contexts of each order $m$ *w.r.t.* each variable $i$ to approximate $\phi_i^{(m)}(x|b)$. Thus, the real computational cost of $L_{\text{Shapley}}$ and $L_{\text{marginal}}$ was reduced to $DK \cdot \sum_{m=0}^{\lambda} T$, where we set $T = 100$ in implementation.

We conducted an experiment to show the time cost of learning the optimal baseline values. We used the MLP, LeNet, and ResNet-20 trained on the *income* dataset, the *credit* dataset, the MNIST dataset, and the CIFAR-10 dataset. For the MNIST and the CIFAR-10 datasets, we randomly sampled $K = 100$ pixels as the variable set $N$ to compute the Loss $L_{\text{Shapley}}$ and optimize their baseline values. Table 12 reports the time cost of optimizing $L_{\text{Shapley}}$ for an epoch *w.r.t.* DNNs trained on different datasets. The time cost was measured by using PyTorch 1.10 on Ubuntu 20.04, with the Intel(R) Core(TM) i9-10900X CPU @ 3.70GHz and one NVIDIA Geforce RTX 3090 GPU.

Table 12: Time cost of learning baseline values for an epoch.

| Model | Dataset | Number of input variables $n$ | Time per epoch | Overall computational time until convergence |
|---|---|---|---|---|
| 5-layer MLP | UCI Census Income dataset | 12 | 3.81s | 1h3min (1000 epochs) |
| 5-layer MLP | UCI South German Credit dataset | 20 | 5.58s | 1h33min (1000 epochs) |
| LeNet | MNIST | 28×28 | 45.69s | 2h32min (200 epochs) |
| ResNet-20 | CIFAR-10 | 32×32 | 64.44s | 3h44min (200 epochs) |

## H  Multi-order Shapley values and marginal benefits

In Section 4 of the main paper, we propose the Shapley values of different orders $\phi_i^{(m)}$, and marginal benefits of different orders $\Delta v_i(S)$. Furthermore, we find that high-order causal effects are only contained by high-order Shapley values and marginal benefits. Actually, the loss function on marginal

benefits $L_{\text{marginal}}$ is more fine-grained than the loss function on the multi-order Shapley value $L_{\text{Shapley}}$. This section provides proofs for the above claims.

First, the Shapley value $\phi_i$ can be decomposed into the sum of Shapley values of different orders $\phi_i^{(m)}$ and marginal benefits of different orders $\Delta v_i(S)$, as follows.

$$\phi_i = \frac{1}{n}\sum_{m=0}^{n-1}\phi_i^{(m)} = \frac{1}{n}\sum_{m=0}^{n-1}\mathbb{E}_{S\subseteq N\setminus\{i\},|S|=m}\Delta v_i(S) \tag{9}$$

where the Shapley value of $m$-order $\phi_i^{(m)} \overset{\text{def}}{=} \mathbb{E}_{S\subseteq N\setminus\{i\}|S|=m}\left[v(\boldsymbol{x}_{S\cup\{i\}}) - v(\boldsymbol{x}_S)\right]$, and the marginal benefit $\Delta v_i(S) \overset{\text{def}}{=} v(\boldsymbol{x}_{S\cup\{i\}}) - v(\boldsymbol{x}_S)$.

• *Proof*:

$$
\begin{aligned}
\phi_i &= \sum_{S\subseteq N}\frac{|S|!(n-1-|S|)!}{n!}\left[v(\boldsymbol{x}_{S\cup\{i\}}) - v(\boldsymbol{x}_S)\right] \\
&= \sum_{m=0}^{n-1}\sum_{S\subseteq N,|S|=m}\frac{|S|!(n-1-|S|)!}{n!}\left[v(\boldsymbol{x}_{S\cup\{i\}}) - v(\boldsymbol{x}_S)\right] \\
&= \frac{1}{n}\sum_{m=0}^{n-1}\sum_{S\subseteq N,|S|=m}\frac{|S|!(n-1-|S|)!}{(n-1)!}\left[v(\boldsymbol{x}_{S\cup\{i\}}) - v(\boldsymbol{x}_S)\right] \\
&= \frac{1}{n}\sum_{m=0}^{n-1}\mathbb{E}_{S\subseteq N,|S|=m}\left[v(\boldsymbol{x}_{S\cup\{i\}}) - v(\boldsymbol{x}_S)\right] \\
&= \frac{1}{n}\sum_{m=0}^{n-1}\phi_i^{(m)} \\
&= \frac{1}{n}\sum_{m=0}^{n-1}\mathbb{E}_{S\subseteq N\setminus\{i\},|S|=m}\Delta v_i(S)
\end{aligned}
$$

**Connection between multi-variate interactions and multi-order marginal benefits.** The $m$-order marginal benefit can be decomposed as the sum of multi-variate interaction benefits, as follows. Therefore, high-order causal effects $U_S$ are only contained in high-order marginal benefits.

$$\Delta v_i(S) = \sum_{L\subseteq S}U_{L\cup\{i\}} \tag{10}$$

• *Proof*:

$$
\begin{aligned}
\text{right} &= \sum_{L\subseteq S}U_{L\cup\{i\}} \\
&= \sum_{L\subseteq S}\left[\sum_{K\subseteq L}(-1)^{|L|+1-|K|}v(\boldsymbol{x}_K) + \sum_{K\subseteq L}(-1)^{|L|-|K|}v(\boldsymbol{x}_{K\cup\{i\}})\right] \\
&= \sum_{L\subseteq S}\sum_{K\subseteq L}(-1)^{|L|-|K|}\left[v(\boldsymbol{x}_{K\cup\{i\}}) - v(\boldsymbol{x}_K)\right] \\
&= \sum_{L\subseteq S}\sum_{K\subseteq L}(-1)^{|L|-|K|}\Delta v_i(K) \\
&= \sum_{K\subseteq S}\sum_{P\subseteq S\setminus K}(-1)^{|P|}\Delta v_i(K) \qquad \text{\% Let } P = L\setminus K \\
&= \sum_{K\subseteq S}\left(\sum_{p=0}^{|S|-|K|}\binom{|S|-|K|}{p}(-1)^p\right)\Delta v_i(K) \qquad \text{\% Let } p = |P| \\
&= \sum_{K\subseteq S}\left[(1+(-1)^{|S|-|K|})\right]\Delta v_i(K) \qquad \text{\% Let } p = |P|
\end{aligned}
$$

$$= \sum_{K \subsetneq S} 0 \cdot \Delta v_i(K) + \sum_{K=S} \left( \sum_{p=0}^{|S|-|K|} \binom{|S|-|K|}{p}(-1)^p \right) \Delta v_i(K)$$

$$= \Delta v_i(S) = \text{left}$$

**Connection between multi-order interactions and multi-order Shapley values.** The $m$-order Shapley value can also be decomposed as the sum of interaction benefits, as follows. Therefore, high-order causal effects $U_S$ are only contained in high-order Shapley values.

$$\phi_i^{(m)} = \mathbb{E}_{\substack{S \subseteq N \setminus \{i\} \\ |S|=m}} \left[ \sum_{L \subseteq S} U_{L \cup \{i\}} \right] \tag{11}$$

● *Proof*:

$$\phi_i^{(m)} = \mathbb{E}_{S \subseteq N, |S|=m} \Delta v_i(S)$$

$$= \mathbb{E}_{S \subseteq N, |S|=m} \left[ \sum_{L \subseteq S} U_{L \cup \{i\}} \right]$$

$$= \mathbb{E}_{S \subseteq N, |S|=m} \left[ \sum_{L \subseteq S} U_{L \cup \{i\}} \right]$$

