# OpenReview forum: "Can We Faithfully Represent Absence States to Compute Shapley Values on a DNN?"
_ICLR.cc/2023/Conference — ICLR 2023 poster_

### Official Review · Reviewer_5Nqt · 2022-10-23

**Confidence:** 4
**Correctness:** 3
**Technical Novelty And Significance:** 3
**Empirical Novelty And Significance:** 3
**Recommendation:** 6

**Clarity, Quality, Novelty And Reproducibility:**

It is interesting to study interpretable ML; For reproducibility, experimental details need further description.

**Strength And Weaknesses:**

Strong Points:
[1]It is a challenging and crucial task that how to represent the absence of input variables and verify the faithfulness of baseline values of Shapley value for interpretable ML.
[2]The approach of estimating optimal baseline values for Shapley values can ensure the trustworthiness of the attribution.

Weak Points:
[1]The authors did not properly back up their claims with evidence somewhere. For instance, “However, we find that most existing masking methods are not satisfactory from this perspective.” and “However, empirically, this method actually introduces additional information to the input.” which lacks key references to support the claim.
[2]In this paper, how to initialize the  in Eq. (2) which is used to mask variables to represent the absence is not report.
[3]How to evaluate the accuracy of Shapley value? There need more details about the metric.
[4]The computational time and details of software environment are not reported, especially the time cost or complexity of computing base value via the approximation-yet-efficient solution.
[5]Many of the reference should be further improved. For example, few references miss page. In addition, the authors may try to discuss the existing work in published papers (rather than a number of preprinted references from arxiv).



**Summary Of The Paper:**

Aiming at investigating how to represent the absence of input variables and verify the faithfulness of baseline values, this paper proposes to use causal patterns to examine whether the masking method faithfully removes information encoded in input variables and a method to learn the optimal baseline value. Experimental results have demonstrated the effectiveness of proposed method.

**Summary Of The Review:**

The absence of input variables and the faithfulness of baseline values of Shapley value are good points.

---

> ### Author Response · Authors · 2022-11-16
> **Response to Reviewer 5Nqt (part 1)**
>
> Thank you very much for your positive feedback and valuable comments.  We will try our best to answer all your questions.
>
> Please let us know if you still have further concerns, or if you are not satisfied with the
> current responses, so that we can further update the response ASAP.
>
> Q1: **Ask to clarify evidence of some claims.** "The authors did not properly back up their claims with evidence somewhere. For instance, 'However, we find that most existing masking methods are not satisfactory from this perspective.' and 'However, empirically, this method actually introduces additional information to the input.' which lacks key references to support the claim.
>
> A1: Thank you. We have followed your suggestions to revise the paper to clarify the experimental evidence for these claims right after the introduction of these claims. Please see the fifth paragraph on Page 2 and the third paragraph on Page 5.
>
> The first claim is that "most existing masking methods are not satisfactory from this perspective," *i.e.,* most existing masking methods cannot remove original causal patterns in the input while not introducing new causal patterns. This claim is supported by experiments in Section 2.1 and Table 1. In Section 2.1, we have conducted experiments to examine whether the existing masking methods (including zero baseline, mean baseline, blur baseline, and conditional baseline) could remove all causal patterns in the original input while not introducing new causal patterns. Results in Table 1 show that all the previous baseline methods did not remove most existing patterns and/or triggered new patterns. Although the baseline value based on the conditional distribution (Frye et al., 2021) performed better than other methods, it still could not be considered an optimal baseline value, according to the results in Table 1. To this end, the learned baseline values by our methods removed much more existing causal patterns and introduced much fewer new causal patterns. We have followed your suggestion to clarify this justification on Page 2 of the main paper.
>
> Besides, the second claim that "this method actually introduces additional information to the input" is actually the same as the first claim, *i.e.*, using the mean value as baseline values may introduce new information (causal patterns). Therefore, this claim is also supported by experimental results in Table 1. Table 1 shows that when using the mean baseline values, the ratio $R$ of the remained/newly introduced patterns was large. This proved that the mean baseline value could not remove all causal patterns in the original input, and it introduced many new causal patterns. In fact, this experimental result was also in line with intuition. It was because, in images, mean baseline values usually introduced many grey dots to the input, as shown in Fig.1. These grey dots formed certain shapes (such as edges and squares), which brought additional information to the input. Nevertheless, we have followed your suggestion to clarify this justification on Page 5 of the main paper.

---

> ### Author Response · Authors · 2022-11-16
> **Response to Reviewer 5Nqt (part 2)**
>
> Q2: **Ask to clarify the initialization of baseline values.** "how to initialize the in Eq. (2) which is used to mask variables to represent the absence is not reported."
>
> A2: Thank you. I suppose this question is about the initialization of baseline values in Eq. (2). In fact, for each specific model, we have repetitively conducted multiple experiments to test the proposed method under different initializations of baseline values, which show that most baseline values learned by our method could stably converged to optimal values, no matter how we initialized baseline values. We have clarified these settings right after Eq. (2) on Page 3, the second paragraph in Section 4.1, and the second paragraph in Section 4.2.
>
> First, for synthetic functions and functions in (Tsang et al., 2018) used in Section 4.1, we learned three sets of baseline values by applying three different initializations of baseline values. Because all input variables in these functions were binary, we initialized baseline values as  0, 0.5, and 1, respectively, to verify the effectiveness of our methods under different initializations. Table 4 in Section 4.1 shows that baseline values learned with different initialization settings all converged to similar and high accuracy, which proved the stability of our method.
>
> Second, for each model learned on each benchmark dataset in Section 4.1, we learned baseline values by initializing baseline values in two different ways. The first initialization was to set initial baseline values to zero (Ancona et al., 2019).  The second initialization was to set initial baseline values to the mean values over different samples (Dabkowski and Gal, 2017). These settings have been introduced in the first paragraph of Section 4.1. Nevertheless, we have followed your suggestion to clarify these settings right after Eq. (2). Figure 3(left) in Section 4.2 shows that baseline values learned with different initialization settings all converged to similar baseline values, except for very few dimensions having multiple local-minimum solutions, which proved the stability of our method.
>
> ---
>
> Q3: **Ask to clarify the metric used to measure the accuracy of Shapley values.** "How to evaluate the accuracy of Shapley value? There need more details about the metric."
>
> A3: Thank you. We have followed your suggestion to provide more details about the metric. Please see the first paragraph on Page 8.
>
> Specifically, in order to design a metric to evaluate the accuracy of Shapley values, we used two tasks where the ground-truth Shapley values could be easily determined. The first task was to explain the model in the Addition-Multiplication dataset proposed by (Zhang et al., 2021), where each model explicitly used causal patterns with pre-determined causal effects for inference. According to Theorem 2, the Shapley value was a uniform allocation of causal effects of each causal pattern to its compositional variables. Therefore, in this task, the ground-truth Shapley values could be easily determined based on causal patterns. Please see Appendix E.6 for more details about this task. The second task was to explain a manually designed decision tree on the MNIST dataset, whose structure directly provided the ground-truth Shapley value for each input variable. Please see Appendix E.13 for more details.
>
> Therefore, we computed the following metric to evaluate the accuracy of the estimated Shapley values. Given an input $x$ with $n$ variables $N=\\{1,2,\ldots,n\\}$, let $\hat\phi_i$ denote the ground-truth Shapley value of the variable $i\in N$, and let $\phi_i$ denote the estimated Shapley value of the variable $i$. If $|\phi_i-\hat\phi_i|\le 0.01$, then we considered the estimated Shapley value $\phi_i$ of the variable $i$ correct; otherwise, incorrect. In this way, the accuracy of the estimated Shapley values was computed as the ratio of input variables with correct estimated Shapley values, *i.e.,*  $\frac{|\\{i\in N||\phi_i-\hat\phi_i|\le 0.01\\}|}{|N|}$. Results in Table 5 and Table 7 show that our method exhibited the highest accuracy of Shapley value on both synthetic functions and image datasets, proving the correctness of baseline values learned by our method.

---

> ### Author Response · Authors · 2022-11-16
> **Response to Reviewer 5Nqt (part 3)**
>
> Q4: **Ask to clarify the computational cost of the algorithm.** "The computational time and details of software environment are not reported, especially the time cost or complexity of computing base value via the approximation-yet-efficient solution."
>
> A4: Thank you. We have followed your suggestion to discuss more about the theoretical computation cost of the proposed method and **conduct new experiments** to evaluate the real computation complexity in real applications. Specifically, in Appendix F.15, we introduce both the theoretical complexity of the loss function and the real complexity of approximating the loss in real applications.
>
> In terms of the theoretical complexity, the complexity of the loss functions $L_{Shapley}$ and $L_{marginal}$ in Eq.(8) is $Dn\cdot \sum_{m=0}^\lambda \binom{n}{m}$, where $D$ denotes the number of samples $\boldsymbol{x}$ in the dataset. $n$ is the number of input variables in the input. $\sum_{m=0}^\lambda\binom{n}{m}$ denotes the computational complexity of all terms $\phi_i^{(m)}(x|b)$ of different orders $m\le\lambda$.
>
> However, as introduced in the second paragraph on Page 7, we applied an approximation method to speed up the computation in real implementation. Specifically, we randomly sampled $K$ input variables in each epoch, and we randomly sampled $T$ contexts of each order $m$ *w.r.t.* each variable $i$ to approximate $\phi_i^{(m)}(x|b)$. Thus, the real computational cost of $L_{Shapley}$ and $L_{marginal}$ was reduced to $DK\cdot \sum_{m=0}^\lambda T$, where we set $T=100$ in implementation.
>
> Finally, we have followed your suggestion to **conduct a new experiment** to show the time cost of learning baseline values. We used the MLP, LeNet, and ResNet-20 trained on the *income* dataset, the *credit* dataset, the MNIST dataset, and the CIFAR-10 dataset. For the MNIST and the CIFAR-10 datasets, we randomly sampled $K=100$ pixels as the variable set $N$ to compute the loss $L_{Shapley}$ and optimize their baseline values. The following table reports the time cost of optimizing $L_{Shapley}$ for an epoch *w.r.t.* DNNs trained on different datasets. The time cost was measured by using PyTorch 1.10 on Ubuntu 20.04, with the Intel(R) Core(TM) i9-10900X CPU @ 3.70GHz and one NVIDIA Geforce RTX 3090 GPU.
>
> | Model       | Dataset                         | Number of input variables $n$ | Time per epoch | Overall computational time of learning baseline values until convergence |
> | ----------- | ------------------------------- | ----------------------------- | -------------- | ------------------------------------------------------------ |
> | 5-layer MLP | UCI Census Income dataset       | 12                            | 3.81s          | 1h3min (1000 epochs)                                         |
> | 5-layer MLP | UCI South German Credit dataset | 20                            | 5.58s          | 1h33min (1000 epochs)                                        |
> | LeNet       | MNIST                           | 28*28                         | 45.69s         | 2h32min (200 epochs)                                         |
> | ResNet-20   | CIFAR-10                        | 32*32                         | 67.44s         | 3h44min (200 epochs)                                         |
>
> ---
>
> Q5: **About the reference.** "Many of the reference should be further improved. For example, few references miss page. In addition, the authors may try to discuss the existing work in published papers (rather than a number of preprinted references from arxiv)."
>
> A5: Thank you. We have followed your suggestions to complete the information (such as the page range) in the reference. Moreover, we have followed your suggestion to use the reference of the published version of papers, instead of the arXiv version. Please see the reference section.

---

### Official Review · Reviewer_bMFX · 2022-10-28

**Confidence:** 4
**Correctness:** 3
**Technical Novelty And Significance:** 3
**Empirical Novelty And Significance:** Not applicable
**Recommendation:** 6

**Clarity, Quality, Novelty And Reproducibility:**

- It is in general quite easy to follow the logic of the authors.
- The idea, to my knowledge, is novel. Also authors have addressed previous work that tried to tackle the problem, such as conditional distribution.
- The experiments are reproducible with details.



**Strength And Weaknesses:**

- Strength:
1. The paper proposes a novel idea to find the optimal baseline values in feature attribution. Empirically it achieves very good results compared to other existing methods such as zero/mean baselines.
- A side weakness here: Why was conditional method not compared for CIFAR-10?

- Theoretical weakness:
1. The authors propose their criteria of choosing baseline values based on the observation that " incorrect baseline values generate
new causal pattern". While this observation is quite intuitive, why is the converse true? That is, it is a bit hard to see why minimizing the number of causal patterns would provide a good baseline value.
- Clarity weakness:
2. A minor point: Theorem 3 needs to be more concise. Is it possible to move part of it into a different property, or remark, or derivations ahead of the formal theorem?
- Complexity of the algorithm: There is a trade-off between complexity and accuracy when we approximate the optimization. In practice, the proposed algorithm could be computationally intensive to get good baseline values, in particular for complex problems with higher orders of interactions.

**Summary Of The Paper:**

The paper proposes a novel idea based on minimizing causal patterns to find the optimal baseline values in feature attribution.

**Summary Of The Review:**

The paper proposes novel ideas with relatively weak theoretical support, but good empirical results. The proposed algorithm can be computationally intensive.

---

> ### Author Response · Authors · 2022-11-16
> **Response to Reviewer bMFX (part 1)**
>
> Thank you very much for your appreciation of the contribution and novelty of this paper. We will try our best to answer all your concerns.
>
> Please let us know if you still have further concerns, or if you are not satisfied with the current responses, so that we can further update the response ASAP.
>
> Q1: **Ask for comparing conditional method on the CIFAR-10 dataset.**
>
> A1: Thank you. We have followed your suggestion to **conduct a new experiment** to compare the conditional method (Frye et al., 2021) with our method on the CIFAR-10 dataset. Please see Table 1 for the result. We first learned a generative model on the CIFAR-10 dataset in the same way as the one trained on the MNIST dataset in the paper, so as to compute the conditional distribution $p(x'|x_S)$. We took samples $x'$ with a large probability $p(x'|x_S)$ as baseline values according to the method in (Frye et al., 2021).
>
> We computed the ratio $R^{(\text{conditional})}$ of the remaining/newly introduced patterns, which were generated by the conditional method. Then, we compared $R^{(\text{conditional})}$ with the ratio $R^{(\text{ours})}$ generated by our method. The following table shows that our method exhibited the lowest value $R^{(\text{ours})}=0.1211$. The conditional method generated a smaller value of the ratio $R$ than the zero baseline values, the mean baseline values, and the blurring baseline values, but the conditional method still generated a larger value of $R$ than our method. It means that although the conditional method performed better than other existing baseline values in terms of removing original causal patterns without introducing new causal patterns, the conditional method still could not generate optimal baseline values.
>
> |                  | $R^{(\text{zero})}$ | $R^{(\text{mean})}$ | $R^{(\text{blur})}$ | $R^{(\text{conditional})}$ | $R^{(\text{our})}$ |
> | ---------------- | ------------------- | ------------------- | ------------------- | -------------------------- | ------------------ |
> | CIFAR-10 dataset | 0.6630              | 0.8042              | 0.7288              | 0.4027                     | **0.1211**         |
>
> ---
>
> Q2: The observation that "incorrect baseline values generate new causal pattern" is quite intuitive.
>
> A2: Thank you. We have followed your suggestion to provide theoretical proof for the claim that "incorrect baseline values generate new causal patterns" on Page 4. In Remark 2, we prove that correct baseline values can correctly explain the sparse representation of causal patterns in the model. In Theorem 3, we further prove that when using incorrect baseline values, a causal pattern will be mistakenly explained as a large number of causal patterns, which significantly increases the complexity of the explanation. Theorem 4 further proves that incorrect baseline values lead to the emergence of a large number of low-order causal patterns. In this way, we theoretically prove that incorrect baseline values mistakenly generate a large number of new causal patterns.
>
> ---
>
> Q3: "... why is the converse true? That is, it is a bit hard to see why minimizing the number of causal patterns would provide a good baseline value."
>
> A3: This is a good question. We have followed your suggestion to discuss this problem in Section 2.1 and Section 2.2. We would like to clarify this problem from two perspectives.
>
> The first perspective is based on the theoretical proof in Theorem 3. Theorem 3 proves that correct baseline values will generate *the least causal patterns*. However, incorrect baseline values will significantly increase the number of causal patterns. Therefore, minimizing the number of causal patterns can avoid the risk of using incorrect baseline values.
>
> The second perspective is to learn baseline values that achieve the simplest explanation of a DNN. According to Occam's Razor, we consider that the simplest causality explanation is the most likely to represent the true inference logic of a DNN. From this perspective, we iteratively revise baseline values to minimize the number of salient causal patterns, so as to achieve the simplest causality (*i.e.*, achieving the simplest causality), and we consider such baseline values as the optimal baseline values.

---

> ### Author Response · Authors · 2022-11-16
> **Response to Reviewer bMFX (part 2)**
>
> Q4: **Ask to make Theorem 3 more concise.** "A minor point: Theorem 3 needs to be more concise. Is it possible to move part of it into a different property, or remark, or derivations ahead of the formal theorem?"
>
> A4: Thank you. We have followed your suggestion to make Theorem 3 more concise. We have split it into Remark 2, Theorem 3, and Theorem 4 in the revised version.
>
> In Remark 2, we prove that correct baseline values can correctly explain the sparse representation of causal patterns in the model.
>
> In Theorem 3, we further prove that when using incorrect baseline values, a causal pattern will be mistakenly explained as a large number of causal patterns, which significantly increases the complexity of the explanation.
>
> Theorem 4 further proves that incorrect baseline values lead to the emergence of a large number of low-order causal patterns. Please see Section 2.1 for details.
>
> ---
>
> Q5: **Ask to clarify the computational cost of the algorithm.** “In practice, the proposed algorithm could be computationally intensive to get good baseline values, in particular for complex problems with higher orders of interactions.”
>
> A5:  Thank you. We have followed your suggestion to discuss more about the theoretical computation cost of the proposed method and **conduct new experiments** to evaluate the real computation complexity in real applications. Specifically, in Appendix F.15, we introduce both the theoretical complexity of the loss function and the real complexity of approximating the loss in real applications.
>
> In terms of the theoretical complexity, the complexity of the loss functions $L_{Shapley}$ and $L_{marginal}$ in Eq.(8) is $Dn\cdot \sum_{m=0}^\lambda \binom{n}{m}$, where $D$ denotes the number of samples $\boldsymbol{x}$ in the dataset. $n$ is the number of input variables in the input. $\sum_{m=0}^\lambda\binom{n}{m}$ denotes the computational complexity of all terms $\phi_i^{(m)}(x|b)$ of different orders $m\le\lambda$.
>
> However, as introduced in the second paragraph on Page 7, we applied an approximation method to speed up the computation in real implementation. Specifically, we randomly sampled $K$ input variables in each epoch, and we randomly sampled $T$ contexts of each order $m$ *w.r.t.* each variable $i$ to approximate $\phi_i^{(m)}(x|b)$. Thus, the real computational cost of $L_{Shapley}$ and $L_{marginal}$ was reduced to $DK\cdot \sum_{m=0}^\lambda T$, where we set $T=100$ in implementation.
>
> Finally, we have followed your suggestion to **conduct a new experiment** to show the time cost of learning baseline values. We used the MLP, LeNet, and ResNet-20 trained on the *income* dataset, the *credit* dataset, the MNIST dataset, and the CIFAR-10 dataset. For the MNIST and the CIFAR-10 datasets, we randomly sampled $K=100$ pixels as the variable set $N$ to compute the loss $L_{Shapley}$ and optimize their baseline values. The following table reports the time cost of optimizing $L_{Shapley}$ for an epoch *w.r.t.* DNNs trained on different datasets. The time cost was measured by using PyTorch 1.10 on Ubuntu 20.04, with the Intel(R) Core(TM) i9-10900X CPU @ 3.70GHz and one NVIDIA Geforce RTX 3090 GPU.
>
> | Model       | Dataset                         | Number of input variables $n$ | Time per epoch | Overall computational time of learning baseline values until convergence |
> | ----------- | ------------------------------- | ----------------------------- | -------------- | ------------------------------------------------------------ |
> | 5-layer MLP | UCI Census Income dataset       | 12                            | 3.81s          | 1h3min (1000 epochs)                                         |
> | 5-layer MLP | UCI South German Credit dataset | 20                            | 5.58s          | 1h33min (1000 epochs)                                        |
> | LeNet       | MNIST                           | 28*28                         | 45.69s         | 2h32min (200 epochs)                                         |
> | ResNet-20   | CIFAR-10                        | 32*32                         | 67.44s         | 3h44min (200 epochs)                                         |

---

### Official Review · Reviewer_EX6v · 2022-10-31

**Confidence:** 4
**Correctness:** 3
**Technical Novelty And Significance:** 3
**Empirical Novelty And Significance:** 3
**Recommendation:** 6

**Clarity, Quality, Novelty And Reproducibility:**

- The paper is novel in trying to find the optimal baseline values that remove the causal effect encoded in the masked variables.
- There are some clarification problems stated above.
- The experiments are reproducible.

**Strength And Weaknesses:**

Strength

- The experiments are well-established on different datasets. The performance shows the effectiveness of the proposed method.

- The approximate-yet-efficient solution is nice to reduce the low-order causal patterns and the number of salient causal patterns.

Concerns

- The problem is not defined clearly. Could you clarify the relationship between the absence of states of the input variables and the absence of the causal effect of the input variables at the beginning? It seems that the masking method you seek is actually aiming to remove the causal effect of the masked variable. It's confusing to not have a clear description of the problem right off the bat of what you are to remove by masking.

- The presentation and some clarification issues. Some fundamental concepts need to be clarified. For example, could you please explain what it means by ‘removing causal pattern’ and ‘introducing causal pattern’ when you first mention it? Also, in this paper, the masking method is considered as faithfully representing the absence states of input variables if it faithfully removes old causal patterns without introducing new causal patterns. But first, to make it clearer, could you please present this definition more formerly; second, for me, it is not that straightforward to define it from this perspective. For example, I am confused about why masking $forehand$ should remove the information of the whole pattern $\\{forehand, eyes, beak\\}$. Even the causal effect of $forehand$ can be removed, is there any proof here? Moreover, could you please give some explanation on why the masking method should be considered in this way?  Is there a loss of information except for the masked variable? If yes, is there any metric for such a loss?

- What is the computational complexity of the proposed method to estimate the optimal baseline values?


**Summary Of The Paper:**

This paper examines whether the masking method faithfully removes the information encoded in the input variables. Then the authors propose a method to remove the effect encoded in the input variables by learning optimal baseline values for Shapley values. Experimental results demonstrate the effectiveness of the method.

**Summary Of The Review:**

This paper finds the optimal baseline values that help to remove the causal effect encoded in the masked variables. I do admit the contribution of this paper, but there are some clarification issues that need to be resolved.

---

> ### Author Response · Authors · 2022-11-16
> **Response to Reviewer EX6v (part 1)**
>
> Thank you very much for your appreciation of the novelty of this paper. We will try our best to answer all your concerns.
>
> Please let us know if you still have further concerns, or if you are not satisfied with the current responses, so that we can further update the response ASAP.
>
> Q1: **Ask to clarify the relationship between the absence of input variables and the absence of causal effects**. "The problem is not defined clearly. Could you clarify the relationship between the absence of states of the input variables and the absence of the causal effect of the input variables at the beginning? It seems that the masking method you seek is actually aiming to remove the causal effect of the masked variable. It's confusing to not have a clear description of the problem right off the bat of what you are to remove by masking."
>
> A1: Thank you. We have followed your suggestion to clarify the relationship between the absence of input variables and the absence of the causal effect in the introduction and Section 2. We would like to explain such a relationship from the following two perspectives.
>
> The first perspective is the triggering/non-triggering states of a causal pattern based on its definition, when baseline values have been given. In the computation of a causal effect, the absence of an input variable is implemented by masking this input variable with its baseline value, according to Eq. (2). In this way, for a causal pattern $S$, if any variable $i\in S$ is absent(masked) in the input, then the causal pattern $S$ will be not triggered, *i.e.,* $C_S=0$, according to Eq. (3) and Eq. (4). Then, the DNN's inference score will not receive the causal effect $I_S$, and this is considered the non-triggering state of a causal pattern. Therefore, from this perspective, the masking of an input variable $i$ will cause the non-triggering states of all causal patterns containing $i$ and the removal of causal effects of these patterns. *I.e.* if the variable $i$ is absent, then $A_i=0,$ $\forall S\ni i$,  $C_S=\prod_{j\in S}A_j=0$, and $I_S=C_S \cdot U_S= 0$.
>
> The second perspective is to learn the baseline values that correspond to the simplest explanation, *i.e.*, the baseline values minimizing the redundancy of the explanation based on causal patterns. According to Occam's Razor, we consider the simplest causality with the minimum causal patterns may potentially represent the essence of the DNN's inference logic. More crucially, we have theoretically proven that incorrect baseline values will make the inference logic encoded in the DNN be mistakenly explained as much more redundant causal patterns. From this perspective, we iteratively revise baseline values to minimize the number of salient causal patterns (*i.e.*, achieving the simplest causality), and we consider such baseline values as the optimal baseline values.

---

> ### Author Response · Authors · 2022-11-16
> **Response to Reviewer EX6v (part 2)**
>
> Q2: **About the presentation.**
>
> Q2.1: **Ask to clarify "removing causal pattern" and "introducing causal pattern."** "Some fundamental concepts need to be clarified. For example, could you please explain what it means by ‘removing causal pattern’ and ‘introducing causal pattern’ when you first mention it?"
>
> A2.1: Thank you. We have followed your suggestion to clarify  "removing causal pattern" and "introducing causal pattern"  when we first mentioned them in the second paragraph on Page 2.
>
> We explain "removing causal pattern" and "introducing causal pattern" from two perspectives. The first perspective is the intuition perspective. "removing causal pattern" means that when an input variable is masked, causal patterns containing this variable are supposed to be not triggered, and their causal effects are removed. For example, let us consider an input image with a bird on a black background. If we mask an important image region in the bird head to be black, which is the same as the background, then the head pattern is probably deactivated, and its causal effect is removed from the output.  "Introducing new causal patterns" means that baseline values of the masked variables may form new causal patterns with significant causal effects. For example, in Fig.1(right), baseline values of variables in $S_K$ form a specific shape that is abnormal in the input, *e.g.,* new edges and new dotted textures. Then, the causal pattern $S_K$ will contribute a new, abnormal, and considerable numerical score to the DNN inference. In other words, such baseline values create new causal patterns with considerable causal effects and affect the inference score of the DNN. In this case, masking input variables with incorrect baseline values may introduce new information to the DNN, rather than only removing existing information.
>
> The second perspective is to learn the baseline values that minimize the redundancy of causal patterns and achieve the simplest explanation of a DNN. According to Occam's Razor, we consider the simplest causality with the minimum causal patterns may potentially represent the essence of the DNN's inference logic. More crucially, we have theoretically proven that incorrect baseline values will make the inference logic encoded in the DNN be mistakenly explained as much more redundant causal patterns. From this perspective, we iteratively revise baseline values to minimize the number of salient causal patterns (*i.e.*, achieving the simplest causality), and we consider such baseline values as the optimal baseline values.
>
> ---
>
> Q2.2: **Ask to define optimal baseline values in a former position of the paper.** "the masking method is considered as faithfully representing the absence states of input variables if it faithfully removes old causal patterns without introducing new causal patterns. But first, to make it clearer, could you please present this definition more formerly;"
>
> A2.2: Thank you. We have followed your suggestion to present the definition for a faithful baseline value more formerly. We have put it right after the definition of causal patterns. Please see the first paragraph on Page 4.

---

> ### Author Response · Authors · 2022-11-16
> **Response to Reviewer EX6v (part 3)**
>
> Q2.3: **Ask to clarify effects that are removed by the masked input variables.** "it is not that straightforward to define it from this perspective. For example, I am confused about why masking $\text{forehead}$ should remove the information of the whole pattern $\{\text{forehead, eyes, beak}\}$."
>
> A2.3: Thank you. We have followed your suggestion to clarify this on Page 3. Let me explain the removal of causal effects by clarifying the definition and physical meaning of the Harsanyi dividend (Harsanyi, 1963). Given three input variables $(a=\text{forehead},b=\text{eyes},c=\text{beak})$, there essentially exists 7 Harsanyi dividend terms $U_S$, where $S\in\\{\\{a\\},\\{b\\},\\{c\\},\\{a,b\\},\\{a,c\\},\\{b,c\\},\\{a,b,c\\}\\}$. The overall output can be represented as the sum of all Harsanyi dividends, $output=I_{\\{a\\}}+I_{\\{b\\}}+I_{\\{c\\}}+I_{\\{a,b\\}}+I_{\\{a,c\\}}+I_{\\{b,c\\}}+I_{\\{a,b,c\\}}$. Each Harsanyi dividend term represents the AND relationship (interaction) between a specific set $S$ of input variables. Such an AND relationship can be formulated as a causal pattern $S$. Only if all variables in $S$ are present, the pattern $S$ is activated and makes a causal effect $I_S$ on the network output. Otherwise, once any variable in $S$ is absent/masked, the pattern $S$ will be deactivated and $I_S=0$. Therefore, the causal effect (Harsanyi dividend) $I_S$ measures the contribution of $S$ only when all variables in $S$ are present, but does not include contributions of smaller subsets. For example, $I_{\\{a,b,c\\}}$ measures the contribution of the interaction between three variables to the output. If the variable $a$ is absent/masked, then the pattern $\\{a,b,c\\}$ is deactivated and $I_{\\{a,b,c\\}}=0$. As for other causal patterns like $\\{b,c\\}$, because they do not contain the variable $a$, these patterns will not be deactivated by the masking of $a$. In other words, the absence of variable $a$ will only deactivate causal patterns containing $a$ and remove their causal effects, but will not deactivate causal patterns that do not contain $a$.
>
> ---
>
> Q2.4: **Ask for a new proof.** "Even the causal effect of $forehead$ can be removed, is there any proof here?"
>
> A2.4: Thank you. We have followed your suggestion to **provide new proof** to show that the causal effect can be removed by the absence of input variables. Specifically, we prove that if the input variable $i$ is masked, then $\forall S\ni i,w_S=0$. Please see Appendix E for the proof.
>
> ---
>
> Q2.5: **Ask to clarify the motivation of using causal patterns to analyze the masking method.** "Moreover, could you please give some explanation on why the masking method should be considered in this way?" *i.e.,* why do you use causality or causal patterns to analyze the masking method?
>
> A2.5: This is a good question. We have followed your suggestion to provide more explanation of using causal patterns to examine the masking method in the third paragraph of the introduction section. Specifically, it is widely believed that the learning of DNN is a fitting problem, instead of explicitly formulating causality or modeling symbolic concepts like how graphical models do. However, some recent studies (Ren et al., 2021a; Deng et al., 2022) and many experimental results have surprisingly shown that when the DNN was sufficiently trained, the sparse and symbolic interactive relationships between input variables would emerge, which can be formulated as a specific type of concept, *i.e.,* causal patterns. Although this phenomenon seems counter-intuitive, it does exist in different DNNs trained for various tasks.
>
> We have followed your suggestion to **conduct new experiments** to show the emergence of relatively sparse causal patterns (interactive concepts) in different DNNs. We trained the five-layer MLP, CNN, LSTM, ResNet-32, and VGG-16 on the UCI census income dataset, the UCI  TV news channel commercial detection dataset, the SST-2 dataset, and the MNIST dataset, respectively. Then, we computed the causal effect $I(S)$ of all causal patterns $S\subseteq N$ in each input sample, and Fig. 8 in Appendix F.2 shows the strength $|I(S)|$of causal effects in the descending order. We found that in various DNNs learned on different tasks, most causal effects were close to zero, and only a few patterns had considerable causal effects. These results indicate that when we defined concepts in the DNN from the perspective of causal patterns, it could sparsely represent the inference output of the DNN and faithfully reflect the interactive relationship between input variables.
>
> Therefore, based on both the above experimental observations and Occam's Razor, we believe baseline values that ensure the simplest causality in the DNN can reflect the essence of the DNN's inference logic.
>
> More crucially, we have also proven in Theorem 3 that incorrect baseline values will cause the DNN to be mistakenly explained as unnecessarily redundant causal patterns.

---

> ### Author Response · Authors · 2022-11-16
> **Response to Reviewer EX6v (part 4)**
>
> Q2.6: "Is there a loss of information except for the masked variable? If yes, is there any metric for such a loss?"
>
> A2.6: Of course, yes, and there should be. We have followed your suggestion to clarify this problem in the last paragraph on Page 3. It is because different input variables do not contribute to the DNN output independently and individually. Instead, many studies (Sundararajan et al., 2020; Ren et al., 2021a; [cite1]) have shown that different input variables may interact with each other to form AND relationships, which is formulated as causal patterns in this paper, as the elementary inference patterns. From this perspective, when we mask an input variable, we need not only to remove the independent effect of the masked variable, but also to remove all causal patterns (AND relationships), in which the variable gets involved. Therefore, when we mask an input variable $i$, the proposed causal effect explicitly represents the numerical effect of each causal pattern where the input variable $i$ gets involved.
>
> Let us consider the following toy example. For the function $f=2x_1+x_2+3x_1x_2+2x_2x_3x_4$ and the input $x=[1,1,1,1$], when we mask the variable $x_1$, all patterns containing $x_1$ are supposed to be deactivated, including $S_1=\\{x_1\\}$ and $S_2=\\{x_2,x_3\\}$. Therefore, causal effects $I_{S_1}$ and $I_{S_2}$ of these patterns are removed from the output. Then, the output $f=x_2+2x_2x_3x_4$.
>
> [cite1] Huiqi Deng, Na Zou, Mengnan Du, Weifu Chen, Guocan Feng, Xia Hu. A Unified Taylor Framework for Revisiting Attribution Methods, in AAAI 2021.
>
> ---
>
> Q3: **Ask to clarify the computational complexity of the proposed method.** "What is the computational complexity of the proposed method to estimate the optimal baseline values?"
>
> A3: Thank you. We have followed your suggestion to discuss more about the theoretical computation cost of the proposed method, and we **conduct new experiments** to evaluate the real computation complexity in real applications. Specifically, in Appendix F.15, we introduce both the theoretical complexity of the loss function and the real complexity of approximating the loss in real applications.
>
> In terms of the theoretical complexity, the complexity of the loss functions $L_{Shapley}$ and $L_{marginal}$ in Eq.(8) is $Dn\cdot \sum_{m=0}^\lambda \binom{n}{m}$, where $D$ denotes the number of samples $\boldsymbol{x}$ in the dataset. $n$ is the number of input variables in the input. $\sum_{m=0}^\lambda\binom{n}{m}$ denotes the computational complexity of all terms $\phi_i^{(m)}(x|b)$ of different orders $m\le\lambda$.
>
> However, as introduced in the second paragraph on Page 7, we applied an approximation method to speed up the computation in real implementation. Specifically, we randomly sampled $K$ input variables in each epoch, and we randomly sampled $T$ contexts of each order $m$ *w.r.t.* each variable $i$ to approximate $\phi_i^{(m)}(x|b)$. Thus, the real computational cost of $L_{Shapley}$ and $L_{marginal}$ was reduced to $DK\cdot \sum_{m=0}^\lambda T$, where we set $T=100$ in implementation.
>
> Finally, we have followed your suggestion to **conduct a new experiment** to show the time cost of learning baseline values. We used the MLP, LeNet, and ResNet-20 trained on the *income* dataset, the *credit* dataset, the MNIST dataset, and the CIFAR-10 dataset. For the MNIST and the CIFAR-10 datasets, we randomly sampled $K=100$ pixels as the variable set $N$ to compute the loss $L_{Shapley}$ and optimize their baseline values. The following table reports the time cost of optimizing $L_{Shapley}$ for an epoch *w.r.t.* DNNs trained on different datasets. The time cost was measured by using PyTorch 1.10 on Ubuntu 20.04, with the Intel(R) Core(TM) i9-10900X CPU @ 3.70GHz and one NVIDIA Geforce RTX 3090 GPU.
>
> | Model       | Dataset                         | Number of input variables $n$ | Time per epoch | Overall computational time of learning baseline values until convergence |
> | ----------- | ------------------------------- | ----------------------------- | -------------- | ------------------------------------------------------ |
> | 5-layer MLP | UCI Census Income dataset       | 12                            | 3.81s          | 1h3min (1000 epochs)                                   |
> | 5-layer MLP | UCI South German Credit dataset | 20                            | 5.58s          | 1h33min (1000 epochs)                                  |
> | LeNet       | MNIST                           | 28*28                         | 45.69s         | 2h32min (200 epochs)                                   |
> | ResNet-20   | CIFAR-10                        | 32*32                         | 67.44s         | 3h44min (200 epochs)                                   |

---

> > ### Comment · Reviewer_EX6v · 2022-11-30
> > **Response to authors**
> >
> > Thanks for answering my questions. I appreciate that the authors have answered all questions. I am raising my rating to 6: marginally above the acceptance threshold.

---

### Author Response · Authors · 2022-11-16
**Response to all reviewers**

Thanks for all reviewers' great efforts and comments. We have answered all your questions, revised the manuscript, and **conducted new experiments** as requested.

**Please let us know if you still have further concerns, or if you are not satisfied with the current responses, so that we can further update the response ASAP.**

---

### Author Response · Authors · 2022-11-24
**Looking forward to further discussion**

Dear Reviewers,

Thanks for your valuable comments to help the improvement of our paper.

We would appreciate it if you could let us know if our responses have addressed your concerns and whether you still have any other concerns. We would be happy to do any follow-up discussion or address any additional comments.

Best regards,

Authors

---

### Decision · Program_Chairs · 2023-01-20

**Decision:**

Accept: poster

**Justification For Why Not Higher Score:**

The paper has good technical contributions, but it is not significant enough for a spotlight.

**Justification For Why Not Lower Score:**

The score cannot be lower due to its good technical contribution.

**Metareview: Summary, Strengths And Weaknesses:**

This paper aims to tackle the limitation of existing interpretable deep learning methods, i.e., the mask-based attribution methods cannot faithfully represent the absence of input variables. To this end, the authors propose a causality-based method and build the connection to Shapley value. Both theoretical and empirical studies are provided to show the effectiveness of the proposed method.

Overall, the paper is novel and interesting. However, the presentation of this paper was not good and the reviewers pointed out several major issues. Fortunately, the authors addressed these concerns by rewriting a significant part of the paper. All the reviewers finally are happy with the revision and lean towards accepting this paper. Given the novelty and technical contribution of this paper, I would recommend acceptance of this paper.


**Note From Pc:**

if the above contains the word "oral" or "spotlight" please see: "oral" presentation means -> notable-top-5% and "spotlight" means -> notable-top-25%. As stated in our emails, we are disassociating presentation type from AC recommendations